# Generative Distribution Embeddings:
# Lifting autoencoders to the space of distributions for multiscale representation learning

**Nic Fishman**[1,†]**, Gokul Gowri**[2,†]**, Peng Yin**[3]**, Jonathan Gootenberg**[4,5,6,*]**, and Omar Abudayyeh**[4,5,6,*]

[1]Department of Statistics, Harvard University; [2]Laboratory for Information and Decision Systems, MIT;
[3]Wyss Institute for Biologically Inspired Engineering and Department of Systems Biology, Harvard University;
[4]Center for Virology and Vaccine Research, Beth Israel Deaconess Medical Center, Harvard Medical School;
[5]Dept. of Medicine, Div. of Engineering in Medicine, Brigham and Women's Hospital, Harvard Medical School;
[6]Gene and Cell Therapy Institute, Mass General Brigham
[†]*Equal contribution*: njwfish@gmail.com, gokulg@mit.edu.
[*]*Senior authors jointly supervised this work*: jgootenb@bidmc.harvard.edu, omar@abudayyeh.science.

## Abstract

Many real-world problems require reasoning across multiple scales, demanding models which operate not on single data points, but on entire distributions. We introduce generative distribution embeddings (GDE), a framework that lifts autoencoders to the space of distributions. In GDEs, an encoder acts on *sets* of samples, and the decoder is replaced by a generator which aims to match the input distribution. This framework enables learning representations of distributions by coupling conditional generative models with encoder networks which satisfy a criterion we call distributional invariance. We show that GDEs learn predictive sufficient statistics embedded in the Wasserstein space, such that latent GDE distances approximately recover the $W_2$ distance, and latent interpolation approximately recovers optimal transport trajectories for Gaussian and Gaussian mixture distributions. We systematically benchmark GDEs against existing approaches on synthetic datasets, demonstrating consistently stronger performance. We then apply GDEs to six key problems in computational biology: learning donor-level representations from single-nuclei RNA sequencing data (6M cells), capturing clonal dynamics in lineage-traced RNA sequencing data (150K cells), predicting perturbation effects on transcriptomes (1M cells), predicting perturbation effects on cellular phenotypes (20M single-cell images), designing synthetic yeast promoters (34M sequences), and spatiotemporal modeling of viral protein sequences (1M sequences).

## 1 Introduction

Advancements in science and engineering increasingly depend on our ability to reason across multiple scales: modeling not just individual data points, but entire *populations* those datapoints are drawn from. In applications ranging from single-cell genomics to DNA sequence design, the relevant unit of analysis is not an individual sample (e.g., a single cell), but the distribution from which it is drawn (e.g. the cell state or the patient they were sampled from). These settings are fundamentally *hierarchical*: we observe sets of samples from latent distributions, which themselves are drawn from a meta-distribution. Without directly modeling these distributions, population-level

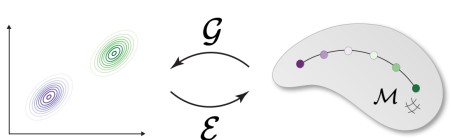

Figure 1: GDEs leverage distribution-invariant encoders ($\mathcal{E}$) and conditional generative models ($\mathcal{G}$) to lift autoencoders to statistical manifolds where points correspond to distributions ($\mathcal{M}$).

signals can be lost underneath unit-level noise. The fundamental challenge we consider is how to learn representations at the level of distributions, not just individual data points.

We introduce *generative distribution embeddings* (GDEs) (Fig. 1), a framework that lifts autoencoders to the distribution space. In GDEs, the encoder maps a finite set of samples – an empirical distribution – to a latent space, while the decoder is replaced by a generative model that reconstructs the distribution by sampling conditional on this latent representation. Our central observation is that strong distributional representations can be learned by coupling conditional generative models with encoders that satisfy a minimal *distributional invariance* property.

Our framework synthesizes modern generative modeling, classical statistics, and information geometry. We show empirically that GDEs behave as approximate *predictive sufficient statistics* [1, 2], capturing distribution-level structure while marginalizing over sampling noise. Moreover, the learned latent spaces exhibit geometric regularity: latent $L_2$ distances correlate with Wasserstein distances ($W_2$) between underlying distributions, and linear interpolation in latent space approximates optimal transport geodesics [3], such that one can generate synthetic data which smoothly interpolate between observed distributions.

We benchmark GDEs on synthetic datasets with known parametric structure, demonstrating improved generative fidelity and structure preservation relative to baselines. We then scale our approach to multiple domains in computational biology, showcasing GDEs' versatility in modeling distributions defined across diverse organizing principles such as distinct populations, varying experimental conditions, spatial arrangements, and temporal dynamics. We demonstrate six applications: learning donor-level representations from single-nuclei RNA sequencing data (6M cells), capturing clonal dynamics in lineage-traced RNA sequencing data (150K cells), predicting perturbation effects on transcriptomes (1M cells), predicting perturbation effects on cellular phenotypes (20M single-cell images), designing synthetic yeast promoters (34M sequences), and spatiotemporal modeling of viral protein sequences (1M sequences). Across these domains, GDEs offer a flexible and scalable framework for distribution-level inference. Code for all experiments is available here.

## 2 Setting and methods

### 2.1 A motivating example

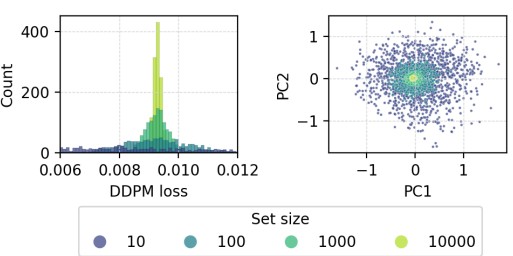

Many modern datasets comprise large groups of *exchangeable* samples: for example, single cells or DNA sequences collected from individual patients [4, 5]. We observe $n$ such groups, each written as $S_{i,m_i} = \{x_{ij}\}_{j=1}^{m_i}$, and collect them as $\mathcal{D} = \{S_{i,m_i}\}_{i=1}^n$ with $x_{ij} \in \mathcal{X}$. Each group's samples are i.i.d. draws from a latent, group-specific distribution $P_i \in \mathcal{P}(\mathcal{X})$, and the $P_i$ themselves are i.i.d. draws from a meta-distribution $Q$; that is, we first sample $P_i \sim Q$ and then, conditional on $P_i$, draw $x_{ij} \sim P_i$. In this framework, a "patient" (or more generally, any group) is implicitly represented by its probability measure $P_i$.

Figure 2: *Concentration of distribution embeddings and plug-in loss. (Left)* Distribution of plug-in GDE loss (diffusion generator) for sets of different sizes sampled from the same distribution $P_i$ over MNIST digits. *(Right)* First two principal components of embeddings of sample sets of different sizes generated by the same $P_i$.

This is a classical hierarchical data generating process, which gives rise to a multiscale problem. The subject of interest here is not only individual cells or DNA sequences $x_{ij}$, but the probability measures $P_i$. As we explore in Sec. 3 and concretely demonstrate in Sec. 6, this setting is broadly applicable beyond the particular case of representing patients as a collection of samples.

In practice, there are often two major challenges in modeling this kind of data. First, unit-level data is often inherently noisy. For example, single-cell data suffers from noise due to molecular undersampling [6]. Our goal is to learn patient embeddings which capture distribution-level signal, rather than the sample-level noise. The second challenge is that groups can contain millions of samples (in the case of DNA sequencing reads per patient, $m$ can be $\sim 10^8$). It is computationally infeasible to train a model on all samples simultaneously, but given the inherent noise at the unit level we would benefit from embedding all available samples at inference time.

In the remainder of Sec. 2, we will show that both of these practical challenges can be overcome by learning distribution embeddings rather than simply encoding sets of samples. The distributional perspective enables models which distill distribution-level signal, and are able to massively scale at inference time to make use of all available data for precise embeddings.

## 2.2 Learning generative distribution embeddings

We address this problem with GDEs, which consist of an encoder $\mathcal{E}$ that maps a finite set of samples $S_{i,m}$ to a latent representation, and a conditional generator $\mathcal{G}$ that (given this representation) induces a distribution on the sample space. Formally, we aim to learn $\mathcal{E}, \mathcal{G}$ such that

$$\mathcal{G}(\mathcal{E}(S_{i,m})) \xrightarrow{m \to \infty} P_i. \tag{1}$$

---
**Algorithm 1** Training GDEs
---
1: **for** each set $S_{i,m_i}$ **do**
2:      Subsample $\tilde{S}_{i,m} \sim S_{i,m_i}$
3:      $z_i \leftarrow \mathcal{E}(\tilde{S}_{i,m})$
4:      $\mathcal{J} \leftarrow \ell(\tilde{S}_{i,m}, \mathcal{G}(z_i))$
5:      Backprop $\mathcal{E}, \mathcal{G}$
6: **end for**
---

The loss $\ell$ is the standard training objective for the conditional generator (for example, an evidence lower bound for a VAE or a denoising score-matching objective for a diffusion model); we do not need to backpropagate through the sampling process of $\mathcal{G}$.

We show that to guarantee Eq. (1) the encoder must satisfy the following two constraints:

1. Permutation invariance: reordering the samples in $S_{i,m}$ does not change the embedding.

2. Proportional invariance: duplicating every sample $K$ times does not change the embedding.

We refer to an encoder with these properties as *distributionally invariant*: the encoder must depend only on the empirical distribution. So for some function $\phi$ we can write:

$$P_{i,m} = \frac{1}{m} \sum_{j=1}^{m} \delta_{x_{ij}}, \qquad \mathcal{E}(S_{i,m}) = \phi(P_{i,m}).$$

We show formally that distributionally invariant encoders can capture any distributional property and furthermore that any non-distributionally invariant architecture can spuriously encode noise features irrelevant to the distribution (formal proofs in App. D.2):

> **Proposition 1.** *(**Informal statement of Corollary 1**) To rule out order and set-size artifacts, the encoder should depend on the sample only through its empirical distribution (equivalently: be permutation invariant and invariant to proportional duplication).*

Beyond separating signal and noise, distributional invariance, coupled with Hadamard differentiability of the pooling operator, has a second consequence: it enables a *central limit theorem for embeddings*. As the set size grows, $\mathcal{E}(S_{i,m})$ concentrates around its population value with Gaussian fluctuations:

$$\sqrt{m}\left(\mathcal{E}(S_{i,m}) - \phi(P_i)\right) \xrightarrow{d} \mathcal{N}(0, \Sigma_{\phi,i})$$

This result, illustrated empirically in Fig. 2, is what makes encoding massive sets possible. The CLT composes through the plug-in loss so that $\ell(S_{i,m}, \mathcal{G}(\mathcal{E}(S_{i,m})))$ is a consistent and asymptotically normal estimator of the population loss. This provides theoretical justification for training GDEs on subsets of larger sample sets: the gradient of the loss computed on small sets matches (in expectation) the gradient computed using all samples per set (see App. D.2):

> **Proposition 2.** *(**Informal statement of Theorem 2**) Fixing $P$, under mild regularity conditions: (i) a distributionally invariant encoder will have asymptotically normal distribution embeddings; (ii) for a suitable divergence, the plug-in loss, $\widehat{\ell}_m = \ell(S_m, \mathcal{G}(\mathcal{E}(S_m)))$ is consistent and asymptotically normal around the population loss; (iii) a global minimizer will recover the true data distribution as $m \to \infty$: $\mathcal{G}(\mathcal{E}(S_m)) \Rightarrow P$. See Fig. 2.*

Violating distributional invariance (for example, by using sum pooling) causes the embedding to depend on set size and breaks this limit theory causing Eq. (1) to fail. In contrast, mean pooling and M/Z-estimators satisfy these properties (see App. D.2).

# 3   From labels to distributions

In the previous section, we focused on a simple motivating example where we have patients and their associated single-cell data. In that setting, the multiscale nature is clear and it is straightforward to define a metadistribution (in other words, how to group samples into sets). In many datasets, a natural hierarchy is not as clear: for example, DNA sequences and expression labels are not multiscale in the same sense as our motivating example.

Here we illustrate how to set up the distribution learning problem in a more general framework based on a dataset of unit-level outcomes associated with labels $D = \{(x_k, y_k)\}_{k=1}^N$ rather than sets drawn from $Q$. We will show how to group data points into sets $\{x_{ij}\}_{j=1}^m$ whose empirical distributions $P_{i,m}$ approximate draws from $Q$. The grouping reflects the structure of the label space $\mathcal{Y}$ and enables us to shape the GDE latent space for downstream applications (see Sec. 5).

When $\mathcal{Y}$ is discrete, we can form sets by grouping datapoints with the same label (e.g., our motivating patient example further explored in Sec. 6.2, cells by clone identity in Sec. 6.3, cell transcriptomes by perturbation in Sec. 6.4, or epigenetic samples by tissue in Sec. B.1). If there is some semantic similarity between discrete labels we can define sets proportional to those similarity metrics, paralleling contrastive learning, where labels define semantic neighborhoods.

When $\mathcal{Y}$ is continuous or structured (e.g. spatial coordinates for the $x_{ij}$ as in Sec. 6.5 or temporal in the viral protein sequences in Sec. 6.7), we can use a similarity kernel to sample points near a target $y_i^*$: $w_{ik} = \exp(-d(y_k, y_i^*)^2/(2\sigma^2))$, defining a probabilistic neighborhood in the label space, enforcing the consideration of the local structure.

When labels are noisy measurements (e.g. expression associated with DNA sequences as in Sec. 6.6), we can invert the noise model $y_k = y_k^* + \epsilon_k$ by sampling a latent target $y_i^*$ and computing likelihood weights $w_{ik} = p(y_k \mid y_i^*)$, yielding samples that reflect the uncertainty of the data.

All these constructions can be unified as instances of a general framework: let $Q^{(\mathcal{Y})}$ be a prior over label distributions. Fix a reference measure $\nu$ on $\mathcal{Y}$ (counting for discrete labels, Lebesgue for continuous) and assume $P_i^{(Y)} \ll \nu$. For each set $i$, we draw $P_i^{(Y)} \sim Q^{(\mathcal{Y})}$ and compute weights

$$\tilde{w}_{ik} \propto \frac{dP_i^{(Y)}}{d\nu}(y_k), \qquad w_{ik} := \frac{\tilde{w}_{ik}}{\sum_{k'=1}^N \tilde{w}_{ik'}},$$

and sample $x_{ij}$ from $D$ accordingly. This framework subsumes the above examples and gives us a general set of tools for shaping the GDE latent space, as we will illustrate in Sec. 6.

# 4   Related work

Several lines of literature have tried to learn distribution embeddings or summary statistics. Kernel methods, such as kernel mean embedding (KME) and set kernels, provide nonparametric approaches to represent probability measures as points in a reproducing kernel Hilbert space, enabling tasks like distributional regression and classification [7, 8, 9, 10, 11]. GDEs naturally nest these methods as particular choices of distributionally invariant encoders. GDEs also generalize the approach in [12], where they develop a particular encoder and VAE-based generator.

Distribution embeddings have also been studied from a geometric perspective. Building on theoretical foundations from Amari [13], several works model distributions as points on a manifold imbued with the Fisher-Rao metric [14, 15, 16, 17]. These methods are either not generative or restricted to categorical distributions. Building on the work of Otto [3], others have considered learning flows over Wasserstein spaces [18, 19] (see Appendix C.2 for background on Wasserstein spaces), primarily focused on leveraging distribution encodings for transport problems as opposed to GDEs which aim to auto-encode distributions. GDEs are complementary to these works, and can be plugged in to many of these frameworks. One recent method closely related to GDEs, Wasserstein Wormhole [20], aims to represent distributions as points in a space where Euclidean distances match Sinkhorn divergences in the sample space. Wasserstein Wormhole is a particular instantiation of a GDE, using an attention-based encoder and generator that only samples a fixed number of points.

A related body of work aims to learn informative summary statistics [21, 22, 23, 24, 25]. These methods typically consider a supervised setting with a particular inferential target. For example, in

the context of likelihood-free inference, one aims to learn summary statistics which are maximally informative about the parameters of a generative model [23, 24].

GDEs are distinct from these approaches along several dimensions: first, we generalize these methods under a common framework with a central objective of re-sampling the encoded distribution (1); second, we develop theory to guide the design and analysis of GDEs toward this objective; third, we show that distribution embedding is deeply related to generative modeling, enabling domain-specific generative models to be bootstrapped into high-quality GDEs to tackle multiscale problems.

On the architectural side, the encoder in the GDE framework requires a distributionally invariant model. While distributional invariance is a concept introduced in this work, it requires permutation invariance, which has been well-studied [26, 27, 28]. Some permutation invariant approaches, such as deep sets [26], are not distributionally invariant due to proportional sensitivity, while others, such as mean-pooled attention layers, are also distributionally invariant (as shown in Appendix D.2).

A key contribution of our work is the observation that any conditional generative model can be repurposed to learn distributional representations. Recent work in the vision domain has found that conditional diffusion models can induce strong image representations [29]. Our work formalizes and generalizes this finding. We demonstrate in practice that a number of modern techniques, including variational autoencoders [30], Sinkhorn-based generative models [31], sliced Wasserstein models [32], denoising diffusion models [33], and autoregressive sequence models [34, 35], can be leveraged to learn GDEs. This is by no means exhaustive: any other conditional generative modelling approach [36, 37], including those which will emerge in the future, can be used in the GDE framework.

## 5 Statistical and geometric properties of GDEs

GDEs aim to learn representations that separate the structure of the data-generating distribution from finite-sample noise, and to synthesize new data consistent with that structure. We formalize this dual role through two complementary perspectives. First, as *predictive sufficient statistics*, GDEs act like learned Rao–Blackwellizations that denoise sampling variability. Second, as *statistical manifold embeddings*, they interpolate between distributions along smooth geometric paths.

### 5.1 Learning an approximate predictive sufficient statistic

The core objective of GDEs is to recover the true data-generating distribution $P$ from finite samples by learning an aggregate representation that distills the structure of $P$ from sampling noise. This objective is captured by the notion of *asymptotic predictive sufficiency* [1, 2], which reformulates sufficiency in terms of conditional independence:

$$\mathbb{P}(x_{\text{new}} \in A \mid T(S_m)) - \mathbb{P}(x_{\text{new}} \in A \mid S_m) \xrightarrow[m \to \infty]{\mathbb{P}} 0,$$

for all measurable $A \subseteq \mathcal{X}$. In our setting, the encoder $\mathcal{E}(S_m)$ serves as such a statistic, asymptotically determining $P$ and marginalizing over sampling variability in $S_m$.

Predictive sufficiency implies a *Rao–Blackwell* improvement principle: conditioning any predictor on a predictively sufficient statistic cannot increase predictive risk under convex loss [2]. To illustrate this empirically, we consider $X_i \sim \text{Pois}(\lambda)$ and predict $\mathbb{P}(X_{n+1} = 0)$. The baseline uses the observed frequency of $X_i = 0$, the GDE estimator conditions on the embedding and draws $10^6$ synthetic samples upon which the baseline estimator is applied, and the Rao–Blackwellized (RB) estimator condi-

| $n$ | Naive | RB | GDE |
|---|---|---|---|
| 10 | 4.72e-03 | 3.79e-03 | 3.12e-03 |
| 100 | 3.41e-04 | 2.76e-04 | 2.71e-04 |
| 1000 | 3.03e-05 | 2.81e-05 | 2.67e-05 |
| 10000 | 3.23e-06 | 2.64e-06 | 3.32e-06 |

Table 1: MSE of Naive, RB, and GDE estimators for $P(X = 0)$, $X \sim \text{Poi}(\lambda)$.

tions on the sufficient statistic $T = \sum_i X_i$. In Tab. 1 we can see that the GDE estimator outperforms the RB bound on the data manifold at low sample numbers. This suggests GDEs can act as data-driven analogues of Rao–Blackwellization, using synthetic sampling to magnify signal.

A predictive sufficient statistic distills the structural properties of the meta-distribution while marginalizing over sampling variability in the observed data. Generative distribution embeddings achieve this in practice: they recover consistent representations of underlying distributions, even across diverse domains and observational sample spaces.

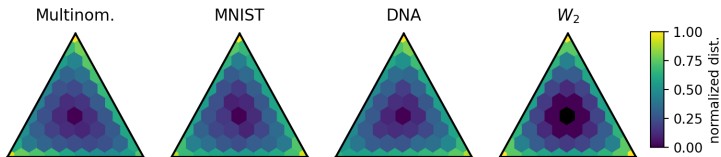

Figure 3: *$L_2$ in GDE latent space compared to $W_2$ distance.* Normalized distances from the center, $p = (1/3, 1/3, 1/3)$. The plots to the left show GDE $L_2$ learned from empirical distributions. MNIST and DNA distributions are constructed by sampling conditional on class label according to a multinomial, for MNIST subsetted to images of (0, 1, 2) and a synthetic DNA dataset with 3 patterns respectively. Rightmost plot shows the Gaussian approximation for the $W_2$ between multinomials.

We demonstrate this using the multinomial distribution. We learn GDEs of 3-dimensional multinomial distributions using a mean-pooled deep sets encoder and a diffusion generator. The model's latent space is able to recover the structure of the multinomial simplex (Fig. 3). Next, we use two real-world datasets with discrete class labels and conditionally sample observations according to label identities, which are drawn from the same family of 3-dimensional multinomial distributions. For both a three-digit subset of MNIST and a set of three synthetic DNA sequence patterns, GDEs (using 2D and 1D convolutional encoders and diffusion and HyenaDNA generators, respectively) recover the same structure of the underlying multinomial simplex in the latent space. Despite coming from three different domains and using three vastly different architectures, the latent geometry learned between these experiments is nearly identical demonstrating GDEs capacity to learn signal from noise.

In fact, the learned geometry is rather particular: the $L_2$ distance in GDE latent spaces in all three cases closely resemble $W_2$ distances between multinomials (computed under a Gaussian approximation). This points to a geometric interpretation of GDEs, bringing us to our second theoretical perspective.

## 5.2 Learning a manifold of distributions

From the geometric perspective, we can try to understand GDEs by examining the structure of their latent spaces. In Fig. 4 we take a "source" and a "target" distribution and compute the corresponding distribution embeddings $z_{\text{src}} = \mathcal{E}(S_{\text{src}})$ and $z_{\text{tgt}} = \mathcal{E}(S_{\text{tgt}})$. We then compute the linear interpolants $z_t = (1-t)z_{\text{src}} + tz_{\text{tgt}}$ and push those back out into distribution space by sampling $\mathcal{G}(z_t)$. The results are rather remarkable: the paths traced in distribution space by the linear latent interpolants closely resemble the optimal transport paths. This motivates a second view of GDEs: they can *generate synthetic distributions* that interpolate smoothly between observed populations. Empirically, latent trajectories in GDE space correspond to families of synthetic data that move coherently through probability space, providing a controllable mechanism for exploring or augmenting realistic generative scenarios.

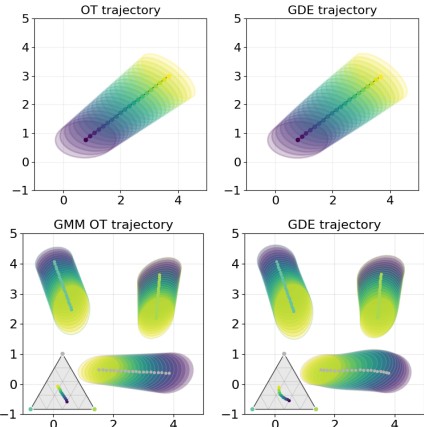

Figure 4: *Top row:* Trajectories between pairs of Gaussians under optimal transport (left) and GDE (right). *Bottom row:* Similar comparison for Gaussian mixture models, we compute the "OT" by finding the optimal pairing between Gaussians and computing the OT. Inset ternary plots show mixture weights during interpolation.

Formally, let $\mathcal{X}$ denote the sample space and $\mathcal{P}_2(\mathcal{X})$ the set of probability measures with finite second moment. The 2-Wasserstein distance $W_2$ makes $\mathcal{P}_2(\mathcal{X})$ a geodesic metric space; in Euclidean settings it also admits a Riemannian interpretation [3]. For a meta-distribution $Q$ over $\mathcal{P}_2(\mathcal{X})$, let $\mathcal{M} \subset \mathcal{P}_2(\mathcal{X})$ be a set supporting $Q$; when $\mathcal{M}$ is a smooth submanifold, it inherits the metric induced by $W_2$. Intrinsic geodesics on $\mathcal{M}$ are shortest paths constrained to lie in $\mathcal{M}$ under this induced metric (and need not coincide with ambient $W_2$ geodesics in $\mathcal{P}_2(\mathcal{X})$).

The encoder $\mathcal{E}$ can be viewed as an (approximate) smooth embedding $\phi : \mathcal{M} \to \mathbb{R}^d$, while the generator parameterizes an approximate inverse that decodes latent trajectories into synthetic distributions along $\mathcal{M}$. Ideally, $\psi$ would preserve the manifold's intrinsic geometry, as an *approximate isometry* that maps Wasserstein distances between distributions to Euclidean distances in latent space. To further assess this, beyond Fig. 4, we can compute the correlation between the latent and Wasserstein distances: for Gaussian distributions, latent-space $L_2$ distances correlate with true $W_2$ distances at $\rho = 0.96$; for 3-component Gaussian mixtures and $W_2$ distances restricted to the mixture family [38], the correlation remains high ($\rho = 0.76$). This highlights a connection between our two perspectives:

> **Proposition 3.** *(Informal statement of Theorem 3) Assume $Q$ is supported on a $d$-dimensional statistical manifold $\mathcal{M}$ and the encoder induces a $C^1$ map $\phi : \mathcal{P}(\mathcal{X}) \to \mathbb{R}^d$ (with the regularity conditions in App. D.3). Then $\phi|_{\mathcal{M}}$ is asymptotically predictively sufficient when $\phi|_{\mathcal{M}}$ is a smooth embedding.*

A key question this exploration prompts is precisely when GDE latent spaces are endowed with Wasserstein geometry. It is worth noting that classically, sufficient statistics and statistical manifolds are fixed once the model family is specified, and the geometry is independent of the likelihood of observing a particular distribution. In contrast, our hierarchical model involves a meta-distributional prior, endowing the setting with a Bayesian flavor. GDEs' predictive sufficiency is therefore evaluated *with respect to $Q$*: favoring statistics that preserve predictive information for distributions that are more common under $Q$. As a result, the learned representation and the synthetic data generated from it become *$Q$-weighted*, allocating resolution to regions of $\mathcal{M}$ according to their probability. This adaptive weighting explains why GDEs can outperform the RB estimator in Tab. 1.

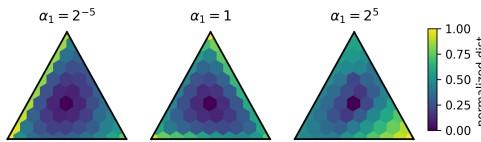

Figure 5: Similar to Fig. 3 we show the GDE distances of multinomials from $p = \left(\frac{1}{3}, \frac{1}{3}, \frac{1}{3}\right)$. We shift the prior asymmetrically by changing $\alpha_1$ while fixing $\alpha_2 = \alpha_3 = 1$. This shifts the focus of the model, leading to a different learned geometry.

In earlier examples $Q$ was approximately uniform over a region of $\mathcal{M}$, yielding a learned geometry close to $\mathcal{P}_2(\mathcal{X})$. When $Q$ is non-uniform, the geometry warps: high-density regions expand to preserve finer distinctions, while low-density regions contract. We demonstrate this in our synthetic multinomial setting by training GDEs on empirical distributions sampled from skewed Dirichlet task distributions, $\alpha = (2^{-5}, 1, 1)$ and $\alpha = (2^5, 1, 1)$, compared to the uniform $\alpha = (1, 1, 1)$ (Fig. 5). As expected, distances stretch precisely where $Q$ concentrates.

The takeaway is operational: the embedding is not only geometrically faithful but also prior-weighted. By choosing $Q$ strategically (e.g., via the task-informed sampling in Sec. 3), we can bias the model toward capturing distributional properties most relevant to downstream objectives.

## 6   Applications

We first benchmark our approach and then demonstrate the generality of GDEs on tasks across the biological sciences, spanning several data domains: DNA sequences, protein sequences, gene expression data, and microscopy data. Throughout, we explore different combinations of encoder-generator pairs, see App. A for a detailed discussion of architectures and training dynamics.

### 6.1   Benchmarking GDEs on synthetic distribution datasets

The design space of GDEs is large: any distributionally invariant encoder can be coupled to any conditional generative model. To guide our implementation choices, we systematically benchmark architectures using synthetic datasets. Included in the benchmarked models are two existing methods that GDEs generalize, kernel mean embeddings and Wasserstein Wormhole [11, 20].

Table 2: Wasserstein reconstruction error across synthetic distributional datasets. Computed as $W_2$ for normal and GMM, and as Sinkhorn divergence for MNIST and FMNIST.

| Model | Normal | GMM | MNIST | FMNIST |
|---|---|---|---|---|
| KME + DDPM | 0.04 | 2.17 | 80.46 | 111.01 |
| $W_2$ Wormhole | 0.20 | 2.88 | 263.29 | 320.18 |
| GDE | **0.02** | **1.82** | **63.79** | **102.21** |

We benchmark 30 combinations of encoders and generators on multivariate normal distributions in 5 dimensions. For evaluation we compute the Wasserstein reconstruction error from ground truth distribution by estimating means and covariance matrices from generated samples and using the closed-form for $W_2$ between Gaussians. We find that mean-pooled deep sets with skip-connections coupled with DDPM generators provide the highest quality generations, outperforming existing techniques. For synthetic distributions we present results for this architecture (see App. B.2).

In Table 2 we additionally benchmark this GDE architecture on three more sophisticated datasets: (1) 3-component Gaussian mixtures in 5 dimensions, (2) mixtures of MNIST [39] images according to categorical distributions of 3 classes, and (3) an analogous dataset using Fashion-MNIST [40]. For image datasets, where $W_2$ distances are not tractable, we instead compute the Sinkhorn divergences between pretrained Resnet18 [41] representations of generated and ground truth samples. In all cases, our chosen GDE architecture outperforms existing approaches.

## 6.2 GDEs enable semi-supervised distribution-level representation learning

We next explore GDEs in our motivating example: for learning patient representations from a single-nucleus RNA-seq atlas of the human prefrontal cortex [4], which profiled over 6.3 million nuclei from 1,494 donors across neurological and psychiatric conditions. We consider each donor's nuclei as samples from an empirical distribu-

Table 3: Patient label prediction from single-cell data. Semi-supervised GDEs improve performance.

| Metric | Supervised | Semi-supervised |
|--------|-----------|-----------------|
| Accuracy | 0.8791 | **0.8887** |
| ROC AUC | 0.4872 | **0.5131** |
| F1 Score | 0.1293 | **0.1479** |

tion, and each condition as a label we wish to predict. As a baseline, we first train a supervised model to predict patient labels from nuclei sets using a mean-pooled deep sets architecture using 10% of the available labelled data. We compare this with a semisupervised model implemented using GDEs. To construct a GDE model, we combine a mean-pooled deep sets encoder with a CVAE generator, and train it using the same 10% labelled data available to the supervised model, along with the remaining 90% with labels withheld. Semi-supervised GDEs outperform supervised baselines across all evaluation metrics (Tab. 3).

## 6.3 Modeling clonal populations in lineage-traced scRNA-seq data

While many methods have been developed for learning representations of single cells from scRNA-seq data [42], methods for learning representations of cell populations remain relatively underexplored. This task is relevant to the analysis of lineage tracing data, where the unit of interest is a *clone*, or a population of cells that arise from the same progenitor.

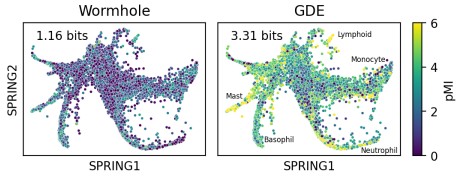

Figure 6: 2D embeddings of lineage-traced scRNA-seq data, hued by pointwise mutual information between clonal representation at early timepoint and clonal fate.

Using lineage-traced scRNA-seq data from mouse hematopoietic stem cells [43], we apply GDEs to learn clone-level representations by treating the set of cells within a clone as samples from an empirical distribution. Following prior frameworks [44], we evaluate the ability of representations to predict future clonal gene expression based on the mutual information (MI) between a clone's representation at an early timepoint during differentiation and its representation at a late timepoint. We find that GDEs with a CVAE generator outperform Wasserstein Wormhole embeddings by over 2 bits (Fig. 6). We next ask if this increase in predictive power is due to improved representations within certain cell types (e.g., neutrophils or monocytes). Decomposing MI estimates into their pointwise contributions [45], reveals contributions across the entire cell state space rather than any particular cell subtype (Fig. 6).

## 6.4 Predicting transcriptional responses to genetic perturbations

A central goal in genomics is to predict the transcriptional effects of genetic perturbations[46, 47, 48]. We evaluate GDE for genetic perturbation prediction, using the Perturb-seq data of Replogle et al. [49] that profiled gene expression responses to CRISPRi knockdown of thousands of genes.

We consider the following task: given the identity of a perturbation, predict the full distribution of transcriptional responses. We compare two approaches. In the first case, we train a linear model to predict the mean expression profile directly. In the second case, we predict the GDE embedding (trained on sets of cells subject to the same perturbation, via a Resnet Deep Sets encoder and CVAE generator as in Sec. 6.3) of the perturbation-induced expression distribution and then recover the mean via a learned linear projection from the embedding space. In both cases, we use a ridge regression on top of GenePT embeddings [50] to enable zero-shot generalization across perturbation conditions, demonstrating that GDE improves both $R^2$ and MSE in Tab. 4. See Appendix B.5 for full details.

## 6.5 Learning morphological cellular responses to genetic perturbations

We apply GDEs to pooled image-based CRISPR screening data from Funk et al. [51], which profiles the phenotypic effects of perturbing 5,072 essential human genes in HeLa cells. The dataset includes over 20 million single-cell microscopy images with four stains, capturing diverse phenotypic variation.

Table 4: GenePT predicting held-out perturbations in mean expression space and GDE latent space.

|  | $R^2$ | MSE |
| --- | --- | --- |
| Mean | 0.378290 | 1.854997 |
| scVI | 0.421491 | 1.551414 |
| GDE | 0.457941 | 1.500731 |

Each perturbation induces a distribution over cell morphologies based on perturbation groupings. We treat these as empirical distributions and train a GDE model to reconstruct them. To explore the role of inductive biases, we instantiate GDEs with two different priors: a spatial prior that models positional image structure (see App. B.6), and a perturbation prior that captures latent variation across perturbation conditions. These approaches capture spatial and perturbation sets, respectively.

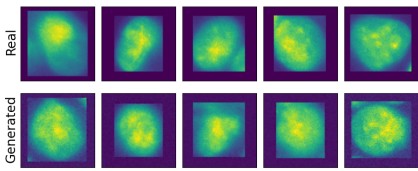

Figure 7: Real/generated DAPI images for the heldout RACGAP1 knockout.

Qualitatively, the model learns to reproduce phenotypic features, including nuclear shape, cytoplasmic texture, and boundary sharpness across perturbations (Fig. 7). Quantitatively, similar to Sec. 6.4 we hold out 30% of the most perturbative perturbations and use ridge regression with GenePT to enable zero-shot generalization across perturbations by predicting the GDE embedding. We then sample conditional on the predicted embedding and compute the nuclear signal intensity. The predictions on these held-out perturbations achieved an $R^2 = 0.7055$ and an MSE of 0.00068, indicating a strong zero-shot generalization of phenotypic outcomes.

## 6.6 Decoding yeast promoter sequence activity with GDEs

We next consider a large-scale dataset from a massively parallel reporter assay measuring transcriptional activity across 34 million randomly generated yeast promoter sequences [52]. Each promoter consists of a random 80 nucleotide DNA sequence embedded in a fixed DNA scaffold and assayed for expression in yeast cells. Because the sequences are randomly sampled, there is no shared structure

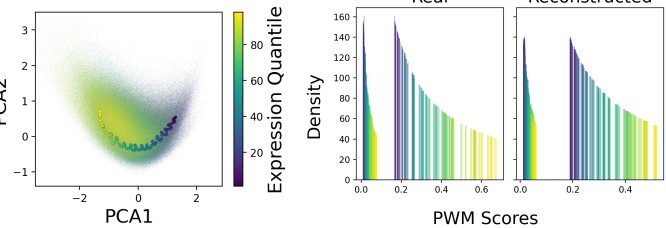

Figure 8: The PCA (left) of the GDE latent space of quantile embeddings with underlying 34 million promoter sequences and the recovered distribution of TFBS (right) as measured by motif counts in both the real and reconstructed data.

across examples so unconditional generative models cannot learn anything meaningful. Instead, the signal lies entirely in how distributions over sequences give rise to distributions over expression levels, due to the presence of transcription factor binding sites (TFBS): short, position-specific DNA motifs that interact with transcription factors and control gene expression [52].

We construct a distributional learning task where each training example is a set of sequences sampled from a narrow expression quantile; we hold out the top 5 quantiles. We train a GDE with a 1D convolutional network over the one-hot encoded sequences as the encoder and HyenaDNA [35] as

the decoder. As shown in Fig. 8, the learned GDE embeddings reflect a smooth gradient across expression quantiles. Using the set of all known yeast TFBS [53] we can identify the motifs present in each of the real and generated sequences. Reconstructed motif distributions closely match those of the input, indicating that the model learns to represent biologically meaningful variation across promoter sets. Further details are available in App. B.8.

### 6.7 Modeling spatiotemporal distributions of viral lineages

Powerful modeling approaches have been developed to represent individual protein sequences [54, 55, 56, 57, 34]. Here, we show that the GDE framework can naturally lift these modeling approaches to learn representations of distributions of sequences. In particular, we model distributions of SARS-CoV2 spike protein sequences over time and location. Using a dataset from the Global Initiative on Sharing All Influenza Data (GISAID) [58], we group sequences by sampling month and site location and treat each group as an empirical distribution over protein sequences. We embed these distributions using a GDE which couples the ESM architecture [56] to a mean-pooled deep sets as the encoder and a conditional ProGen2 architecture [34] as the generator.

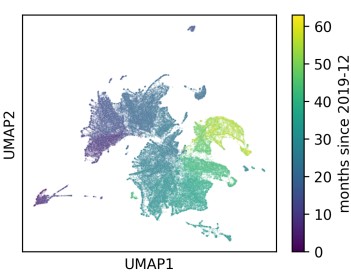

As shown in Fig. 9, the learned latent space organizes samples chronologically, suggesting that GDEs capture time-varying signal about sequence distributions. And indeed, this is observed quantitatively: ridge regression on GDE representations predicts the month of held out sequence distributions with mean absolute error (MAE) of $1.83 \pm 0.01$ months, an improvement over the baseline of mean-pooled ESM embeddings with MAE of $2.24 \pm 0.01$ months (errors reported as mean $\pm$ s.e.m. over 10 random train/test splits). See App B.9 for further details.

Figure 9: GDE representations of protein sequence distributions. Each point corresponds to a set of SARS-CoV2 spike sequences obtained from one lab in one month.

Similarly, we also observe a spatial signal, albeit much weaker. An SVM trained to classify distributions by country achieves $0.28 \pm 0.001$ accuracy from GDE representations, compared to $0.25 \pm 0.003$ from mean-pooled ESM embeddings. Both approaches slightly outperform the baseline of predicting the most common dataset label ('USA' with accuracy $0.21$).

## 7 Discussion

We introduce *generative distribution embeddings*, a framework that couples distribution-invariant encoders with conditional generators to learn structured representations of distributions. Finite sample sets are mapped by smooth embeddings that asymptotically identify the underlying distribution, enabling consistent reconstruction in the large-sample limit. We formalized these properties via connections to predictive sufficiency and statistical manifold embeddings, and proved that a broad class of encoder architectures is asymptotically normal and unbiased when trained via a plug-in loss.

We demonstrated GDEs across a diverse set of large-scale biological problems. These applications highlight the generality of GDEs and their ability to operate directly on measurement data while modeling population-level structure. Crucially, GDEs support flexible distributional constructions (e.g. spatial neighborhoods, time windows, expression quantiles), showing that a wide range of problems can be cast as population-level modeling tasks. Code for model training and dataset preprocessing is available at this Github repository.

**Limitations** GDEs rely on sensible choices of meta-distributional priors (i.e. construction of sets, Sec. 3), often requiring careful, domain-specific design. GDEs also pose practical engineering challenges (propagate gradients to the encoder through the generator, scaling to large set sizes) discussed in App. A. On the theoretical side, the current formalism assumes exchangeable samples, and does not admit non-i.i.d. samples within a distribution. Regarding geometry, we provide empirical but not mechanistic evidence that GDEs learn isometries across domains.

**Extensions** GDEs can serve as a tool for generalization (akin to meta flow matching [19]), can be expanded to settings where the i.i.d. assumption within sets of samples does not hold, and extended to semi-supervised settings. More broadly, GDEs point toward questions at the intersection of empirical process theory, information geometry, and generative modeling; we hope this connection can be explored more deeply in future work.

**Acknowledgements** Funding: J.S.G. and O.O.A. are supported by NIH grants R01-EB031957, R01-AG074932, and R01-GM148745; G. Harold & Leila Y. Mathers Charitable Foundation; Rett Syndrome Research Trust; The Gordon and Betty Moore Foundation; Impetus Grants; Cystic Fibrosis Foundation Pioneer Grant; Google Deepmind; Sanofi; Yosemite; Michelson Foundation; Hevolution Foundation; American Federation for Aging Research; Pivotal Life Sciences; and the MGB Gene and Cell Therapy Institute. G.G. is supported by a Tayebati Fellowship. N.F. is supported by the NSF GRFP.

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

# A    Architectures and Training Dynamics

In this section we outline some general details about the architectures and training dynamics for GDEs. In the following section we will give more detailed explanations about each specific experiment, in addition to full details available in the codebase. All of these findings are somewhat provisional, and there is significant scope for future work to further explore these design choices, but we hope this is a useful complement to our codebase for researchers trying to train their own GDEs.

## A.1    Encoder Architectures

Our framework utilizes permutation-invariant encoders to map input sets $S_m = \{x_1, ..., x_m\}$, where each $x_i \in \mathbb{R}^d$, to a fixed-dimensional latent representation $z \in \mathbb{R}^l$. We primarily employ several types of set encoders, including variants based on self-attention, Graph Neural Network (GNN)-style pooling, and residual connections. All encoders typically conclude by applying a final pooling operation (e.g., mean pooling) across the element representations, followed by a linear projection and a non-linearity (e.g., SELU) to produce the final latent vector $z$.

### A.1.1    Simple Self-Attention Encoder

This encoder provides a baseline transformer-based approach. It first applies a linear layer followed by a SELU activation to project input elements $x_i$ into a hidden dimension $H$. It then processes these representations through a series of multi-head Self-Attention blocks [59]. This architecture directly models pairwise interactions within the set.

### A.1.2    Simple GNN Encoder

The simple GNN-style encoder offers an alternative based on iterative pooling and non-linear transformations, distinct from the standard DeepSets [26] sum-decomposition. It starts with an MLP projection into the hidden dimension $H$. Subsequently, it applies a sequence of layers, each performing a pooling operation across the set followed by an MLP. This structure iteratively refines element representations based on aggregated set information.

**Pooling Operations:** Our theoretical framework (see Appendix D.2) justifies the use of pooling operations that correspond to M/Z-estimators. We focus on mean pooling but additionally implement median pooling as an illustrative example. Notably, max pooling is generally not suitable in this context as its non-differentiability breaks the convergence guarantees we are interested in for Eq. (1), see the remarks in App D.2 for details. Future work might thoroughly explore which pooling operations lead to the greatest flexibility and stability for distribution embedding.

### A.1.3    ResNet-GNN Encoder

To improve gradient flow and enable deeper architectures, we enhance the GNN-style encoder with residual connections. This encoder first projects each input element $x_i$ into $H$ using an MLP. It then processes the set through a series of blocks where each block $k$ computes an intermediate representation $h_i^{(k)}$ for each element $i$. The core operation within a block uses mean pooling (or median pooling). Inspired by ResNet [60, 28], we incorporate skip connections. The input to block $k$ includes the output from the previous block $h^{(k-1)}$, a linear projection of the original input $x$, and the output of the initial MLP projection. Formally:

$$h^{(k)} = \text{LayerNorm}(\text{PooledFC}(h^{(k-1)}) + h^{(k-1)} + \text{Linear}_k(x))$$

where $h^{(0)}$ is the output of the initial input projection combined with a projection of $x$, followed by Layer Normalization. This structure ensures the original input signal is preserved.

### A.1.4    ResNet-Transformer Encoder

This variant follows the same residual structure as the ResNet-MLP encoder but replaces the layers with standard multi-head Self-Attention blocks [59]. This potentially allows the model to learn more complex interactions while benefiting from the improved training dynamics of residual connections. The skip connection mechanism remains identical to the ResNet-MLP version.

### A.1.5 Encoder Comparison

Transformer-based encoders (Simple Self-Attention and ResNet-Transformer) often leverage pre-trained weights effectively and can converge in fewer epochs compared to GNN-style approaches. However, this typically comes at a higher computational cost per epoch and during inference due to the quadratic complexity of self-attention with respect to set size $m$. With sufficient training, we find that the GNN-based architectures, particularly the ResNet-GNN, achieve strong performance, often rivaling the transformer variants while being more computationally efficient for large sets.

Alternative Generative Strategies and Sampling The Wasserstein Wormhole [20] uses a self-attention decoder with fixed positional embeddings that can map the latent $z$ back to samples. One potential method replaces fixed positional embeddings with samples drawn from a simple distribution (e.g., Gaussian) transforming this into a true generator. But this incurs substantial computational costs (e.g., quadratic cost in the number of generated samples for attention-based sampling decoders), and it is not clear this would lead to significant improvements in performance.

It also becomes less obvious how to adapt existing generator architectures using this approach. One option is to use self-attention to construct sample-specific condtional signals from the latent $z$ and the noise vector, and then condition the generator on this signal. This is significantly more complex, and is not clear that this would lead to significant improvements in performance.

## A.2 Adapting Pre-trained Models

Our framework is designed to flexibly incorporate pre-trained models, leveraging their learned representations and generative capabilities. We adapt pre-trained models for both the encoder and the generator components.

### A.2.1 Encoder Adaptation

For tasks involving complex input modalities like natural language or protein sequences, we can utilize pre-trained transformer-based encoders such as BERT [61] or ESM [62] as powerful feature extractors. These pre-trained models can serve as the initial feature extraction layer, whose outputs $\{h_1, ..., h_N\}$ are then fed into the subsequent aggregation layers of our set encoders (e.g., ResNet-GNN or ResNet-Transformer, see subsection A.1).

The adaptation process typically involves:

1. **Loading Pre-trained Weights:** We load the desired pre-trained encoder model using standard libraries like Hugging Faces `transformers` [63].

2. **Feature Extraction:** For each element $x_i$ in the input set $X = \{x_1, ..., x_N\}$, we pass it through the pre-trained transformer to obtain a contextualized representation $h_i$. Often, the output embedding corresponding to a special token (like `[CLS]` in BERT) or the mean/max-pooled output of the final hidden states is used.

3. **Set Aggregation:** These element-wise feature vectors $\{h_1, ..., h_N\}$ are then fed into the subsequent layers of our chosen set encoder (e.g., ResNet-MLP or ResNet-Transformer layers) which perform the permutation-invariant aggregation to produce the final latent representation $z$.

4. **Fine-tuning (Optional):** Depending on the task and dataset size, the pre-trained encoder's weights might be kept frozen initially or fine-tuned jointly with the rest of the model during end-to-end training.

### A.2.2 Generator Adaptation and Conditioning

A core strength of our approach is the ability to use large pre-trained causal language models (LMs), such as GPT-2 [64], ProGen2 [34], or specialized models like HyenaDNA [35], as the conditional generator $p_\theta(x|z)$.

The adaptation involves:

1. **Loading Pre-trained Weights:** We load the chosen pre-trained causal LM and its associated tokenizer using 'transformers' [63].

2. **Prefix Conditioning:** The primary challenge is to effectively condition the generator's output on the latent set representation $z$ produced by the encoder. In practice, we find prefix tuning to be an effective and widely applicable method. The latent vector $z \in \mathbb{R}^L$ is projected, typically via a small MLP $W_p$, into one or more vectors $p = W_p(z)$ that have the same hidden dimension as the LM. These projected vectors $p$ are then treated as continuous "prefix" embeddings prepended to the actual input sequence embeddings $E(x_{<T})$ before they are processed by the transformer layers. The model learns to interpret this prefix as the conditioning signal specifying the target distribution. Mathematically, the input embedding sequence to the transformer becomes $[p; E(x_{<T})]$. The attention mask is adjusted accordingly to allow all sequence tokens $x_{<T}$ to attend to the prefix $p$.

3. **Fine-tuning:** The pre-trained generator weights can be either frozen or fine-tuned. Fine-tuning the entire model allows the LM to adapt its generation process based on the conditioning prefix $p$. Freezing the LM backbone and only training the conditioning projection $W_p$ (and potentially adapter layers) can be more parameter-efficient.

## A.3 Training Details and Considerations

### A.3.1 Learning Rate Schedule

For simpler models we use a fixed learning rate, but for more complex models we typically employ a cosine annealing learning rate schedule during training. This involves starting with an initial learning rate and gradually decreasing it towards zero following a cosine curve over the course of training epochs. This schedule is often effective in achieving stable convergence and good final performance. In general we have found that whatever the current state of the art for training the (unconditional) generator is, that will generally give good results when learning the encoder-generator jointly.

### A.3.2 Performance and Convergence

Our experiments generally indicate that this training setup, combined with the described architectures and adaptation strategies, leads to strong performance across various tasks and datasets presented in the main paper. As noted in subsection A.1.5, the choice of encoder can impact convergence speed and computational cost.

### A.3.3 Set Size and Batching Trade-offs

We observe that achieving optimal performance sometimes necessitates using large input set sizes ($N$). However, processing large sets can significantly increase the computational and memory requirements per batch, particularly for the attention mechanisms in transformer-based encoders or generators. This often forces a reduction in the overall batch size to fit within hardware constraints. Smaller batch sizes can, in turn, lead to increased variance in the loss gradients, potentially slowing down or destabilizing training. Careful tuning of the set size $N$, batch size, and learning rate parameters is often required to balance performance and training efficiency for a given task and hardware setup.

### A.3.4 Gradient Propagation Challenges

A potential challenge arises, particularly with deeper encoder and generator architectures. The encoder only receives a learning signal indirectly through the generator via the shared latent variable $z$. If the generator itself struggles to utilize the latent information effectively, or if the dimensionality $L$ of $z$ creates an information bottleneck, the gradients flowing back to the encoder can become weak or noisy. This can make training deep encoders difficult. Addressing this might require more sophisticated generator architectures capable of integrating the latent information more effectively or alternative training schemes with auxiliary losses directly on the encoder. We found these issues in the simple encoder architectures, but they seemed to be alleviated in the ResNet-based architectures.

## A.4 Implementation recipe for GDEs

The GDE framework is instantiated by pairing a distributionally invariant encoder with a conditional generative model. The following steps outline a general recipe for building GDEs across diverse data domains:

1. **Sample from the metadistribution (construct sets)** Group raw data into sets $S_i = \{x_{ij}\}_{j=1}^{m}$, where each set reflects a draw from an unknown latent distribution $P_i$. Groupings can be based on discrete metadata (e.g., text by author, reviews by rating, images by label, cell clones, gene perturbations) or continuous metadata (e.g., time, location, expression quantiles). Sets need not be mutually exclusive, meaning a single data point can belong to multiple sets.

2. **Choose a distributionally invariant encoder** Select or construct a distributionally invariant encoder $\mathcal{E}$. This selection generally involves (1) using an architecture for element-wise embeddings and (2) pooling across element-wise embeddings with a sample mean (or other M-estimate). We found that architectures with multiple pooling layers, where each layer's pooled output is concatenated with the element-wise embeddings, were particularly effective. This contrasts with pure DeepSets-style architectures that only pool once at the final layer. For deeper architectures, we have found that including skip-connections improves performance, especially if the generator is also a relatively deep network.

3. **Build a conditional generator** The generator $\mathcal{G}$ "decodes" from latent space back to the sample space. It should be conditionable on $z = \mathcal{E}(S)$.

4. **Train via plug-in loss** Optimize the generator to minimize the generator loss function $\ell(P_m, \mathcal{G}(\mathcal{E}(S_m)))$. This loss should be the standard training objective for the conditional generator. This plug-in loss encourages reconstruction of the true distribution.

### A.5  Encoder and Generator Architectures by Experiment

The encoder-generator pairs used for each application in the paper are shown in the table below.

| Sec. | Task | Set Construction | Encoder Arch. | Generator Arch. | Notes |
|------|------|------------------|---------------|-----------------|-------|
| 6.1 | MNIST, FM-NIST | Same image class | see Table 1 | see Table 1 | Synthetic data benchmark |
| 6.2 | Lineage-traced scRNA-seq | Same cell clones | ResNet-GNN | CVAE | |
| 6.3 | Genetic perturbation (scRNA-seq) | Same perturbation | ResNet-GNN | CVAE | |
| 6.4 | Morphological responses (cell images) | Same perturbation | 2D Conv-GNN | DDPM (U-Net) | |
| 6.5 | Tissue-specific methylation | Same patient; Same tissue type | 1D Conv-GNN | HyenaDNA | Uses prefix conditioning |
| 6.6 | Yeast promoter quantile decoding | Expression quantile (continuous) | 1D Conv-GNN | HyenaDNA | Uses prefix conditioning |
| 6.7 | Viral protein spatiotemporal modeling | Same sampling month and location | ESM + mean pooling | ProGen2 | Uses prefix conditioning |

# B Experiments

## B.1 Determining tissue-specific methylation signatures from bisulfite sequencing reads

Analyzing sequencing data typically extensive preprocessing, including alignment to a reference genome. GDEs present an alternative, where sequencing reads can be modeled directly – without alignment or other preprocessing steps. To demonstrate this capability, we show that GDEs can detect tissue-specific DNA methylation patterns directly from bisulfite sequencing (BS-seq) reads. BS-seq measures methylation indirectly through substitution errors: methylated cytosines remain unchanged, while unmethylated cytosines are substituted as thymines. Using publicly available methylation data from diverse tissues [5], we simulate sample-specific BS-seq read distributions by imposing corresponding base substitutions to the reference genome (see Appendix B.7 for details).

Critically, we do not provide the GDE model with any explicit information about methylation signals, the structure of the experimental assay, or a reference genome. The model has access only to sets of sequencing reads grouped by both patient and tissue type. For the GDE model architecture, we choose a 1D convolutional network encoder, and the decoder is a HyenaDNA model [35]. To support large-scale inference over tens of millions of reads per patient, we process 200,000 reads at a time through the encoder and aggregate the resulting embeddings using a simple mean, justified by Theorem 2. This design allows the model to scale efficiently while preserving distributional fidelity.

Our approach enables end-to-end learning of methylation signatures from tissue-specific read distributions. There are two levels of tissue classification, a coarse level with 37 categories and a fine-grained level classification with 83 tissues. Training a linear classifier on top of the GDE latent space, we achieve a test accuracy of 60% on the coarse task and 35% on the fine-grained classification.

## B.2 Additional semi-synthetic experimental results

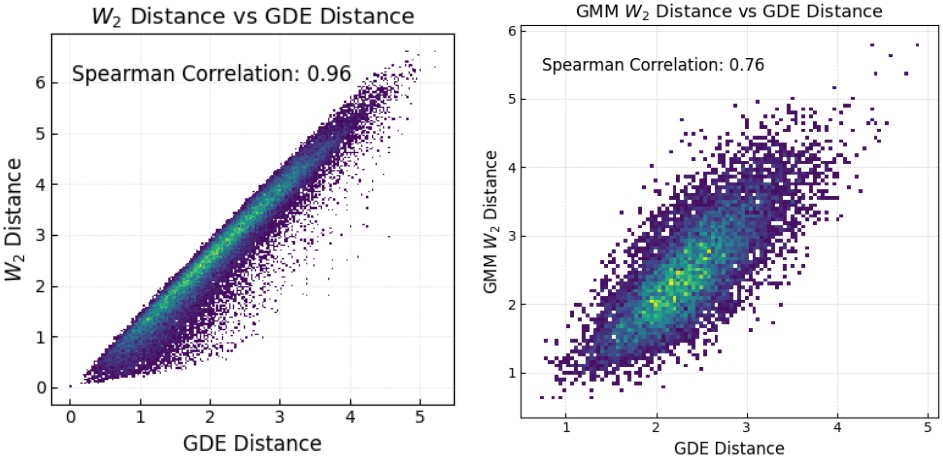

Figure 10: Left: Distance correlation showing high alignment between latent GDE distances and analytical $W_2$ distances (Spearman $\rho = 0.96$). Left: Distance correlation showing high alignment between latent GDE distances and the OT-GMM distance [38], which is a $W_2$ metric restricted to the subspace of GMMs (Spearman $\rho = 0.76$).

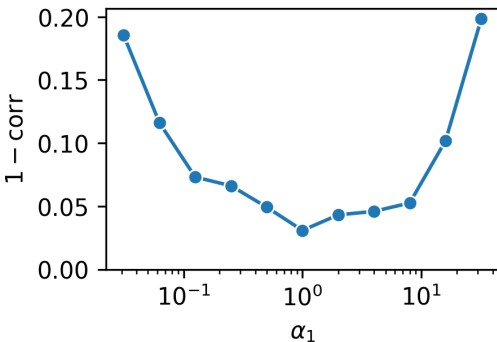

Figure 11: Expanding on Fig. 5 we show that the Pearson correlation between the $W_2$ (computed via normal approximation) and the latent GDE distances decreases as $\alpha_1$ deviates from 1, while keeping fixed $\alpha_2 = \alpha_3 = 1$.

Table 5: $W_2$ reconstruction error of 30 possible GDE implementations (including two existing methods generalized by GDE, Wasserstein Wormhole and kernel mean embeddings) on 5-dimensional multivariate Gaussians. Covariance matrices sampled from Wishart distribution with scale of 1, and means sampled uniformly from $[0, 5]$. Further results included in Table 6.

| Gen. ↓ \\ Enc. → | Mean | Kernel mean | GNN | Med.-GNN | ResNet-GNN | SelfAttn. |
|---|---|---|---|---|---|---|
| Sinkhorn | 0.05 | 0.14 | 0.09 | 0.10 | 0.05 | 0.06 |
| Sliced $W_2$ | 0.03 | 0.04 | 0.07 | 0.07 | 0.03 | 0.04 |
| CVAE | 0.16 | 0.16 | 0.19 | 0.20 | 0.15 | 0.17 |
| DDPM | 0.03 | 0.04 | 0.06 | 0.05 | 0.02 | 0.07 |
| Wormhole | 0.14 | 0.15 | 0.72 | 0.49 | 0.14 | 0.20 |

Table 6: $W_2$ reconstruction error (mean $\pm$ s.e.m. over 5 trials) for 30 possible GDE implementations (including two existing methods generalized by GDE, Wasserstein Wormhole and kernel mean embeddings) on 5-dimensional multivariate Gaussians. Covariance matrices sampled from Wishart distribution with scale of 0.1, and means sampled uniformly from $[0, 5]$.

| Gen. ↓ \\ Enc. → | Kernel mean | GNN | ResNet-GNN | Self-Attn. |
|---|---|---|---|---|
| CVAE | 0.15 ± 0.011 | 0.12 ± 0.006 | 0.12 ± 0.009 | 0.11 ± 0.007 |
| DDPM | 0.15 ± 0.008 | 0.13 ± 0.020 | **0.09 ± 0.003** | 0.10 ± 0.005 |
| Direct SW | 0.15 ± 0.008 | 0.13 ± 0.007 | 0.13 ± 0.009 | 0.15 ± 0.001 |
| Direct Sinkhorn | 0.29 ± 0.008 | 0.22 ± 0.010 | 0.17 ± 0.005 | 0.19 ± 0.010 |
| Wormhole | 0.23 ± 0.021 | 0.72 ± 0.090 | 0.24 ± 0.011 | 0.34 ± 0.021 |

## B.3 Donor-level representation learning experiments

### B.3.1 Data preprocessing

We use single-nucleus RNA-seq data from the Population-scale cross-disorder atlas of the human prefrontal cortex [4], which profiles over 6.3 million nuclei from 1,494 donors across 33 neurological and psychiatric conditions. The dataset consists of multiple sub-datasets, so to avoid integration issues we subset to the largest sub-dataset which contains 4 million cells. For each donor, raw count matrices were normalized to $10^4$ counts per nucleus, log-transformed, and restricted to the top 2,000 highly variable genes. We treat each donor's collection of nuclei as an empirical distribution over transcriptional states. Donor-level diagnostic metadata were obtained from the accompanying PsychAD clinical annotations, and we restrict prediction targets to the six major disease categories which have at least one positive and negative example in the dataset (Alzheimer's, Parkinson's, diffuse Lewy body, bipolar, schizophrenia, and vascular dementia).

### B.3.2 Model architecture and training

Both the supervised and semi-supervised models use the same ResNet deep sets encoder to aggregate single-cell features into donor-level embeddings. For the supervised variant, the encoder is trained end-to-end with a classification head. For the semi-supervised GDE, the same encoder is coupled to a conditional variational autoencoder (CVAE) generator, trained jointly to reconstruct cell distributions while predicting disease labels for the 10% of donors with labeled diagnoses. In both cases, we use a 64-dimensional latent space and two hidden layers of size 128. After training, a logistic regression classifier is fit on the donor embeddings to predict multi-label disease status across the six categories.

### B.3.3 Evaluation

We report donor-level predictive performance using accuracy, balanced accuracy, ROC-AUC, and F1 score on 10% of completely heldout data. Semi-supervised GDEs outperform purely supervised models across all metrics (Table 3), demonstrating that unlabeled donor distributions improve representation quality and generalization in large heterogeneous cohorts.

### B.4 Lineage-traced scRNA-seq experiments

### B.4.1 Data preprocessing details

We use lineage tracing data from Weinreb et al. [43]. The single-cell RNA sequencing (scRNA-seq) count matrices were preprocessed following standard procedures. Specifically, counts for each cell were normalized by rescaling to $10^4$ counts per cell, followed by log transformation. Finally, the top $10^4$ highly variable genes (HVGs) were selected. Cell-type annotations and two-dimensional SPRING embeddings were obtained directly from the annotations provided in Weinreb et al.

### B.4.2 Mutual information estimation

We compute mutual information as a sample mean of pointwise mutual information estimates. To estimate pointwise mutual information in the representation space, we use the nonparametric nearest-neighbor estimator introduced by Kraskov et al. [65] with $k = 3$. This estimator has been shown to be effective in this setting: model latent spaces with tens of dimesions [44].

### B.4.3 GDE modelling architecture

We use a Resnet-GNN architecture as the encoder and a CVAE as the generator. We use 64 latent dimensions, with 2 hidden layers of size 128.

### B.5 Perturbation Prediction

### B.5.1 Data preprocessing details

We use the pre-processed h5ad file from [49] including $10^4$ genes. We compute the 10% most perturbative perturbations by examining the differentially expressed genes and then randomly select 20 of those perturbations to hold out. We hold these out across all cell types.

### B.5.2 GDE modelling architecture

We use a Resnet-GNN architecture as the encoder and a CVAE as the generator, similar to the architecture in the lineage-tracing experiment, except we use a larger hidden state (1024) and a larger latent space (256). We include a perturbation prediction loss during training which trains a linear model with pairwise interactions between the control cell distribution embedding and the gene embedding to predict the difference in mean expression through a linear head. This structures the latent space for our downstream perturbation prediction task.

### B.5.3 Perturbation Prediction

We fit a ridge regression to predict (1) the difference in mean expression and (2) the difference between the perturbed embedding and the control for each perturbation using GenePT gene embeddings [50] with cross-validation to perform grid search over $\lambda$. We then compute the predictions on the held-out perturbations and use a linear head to predict the mean expression from the latent difference. Finally we compute the $R^2$ score and the MSE.

### B.6 Optical pooled screening dataset

#### B.6.1 Data preprocessing details

We use phenotyping images with assigned perturbation barcodes from Funk et al. [51]. We analyze only two of the measured channels: DAPI and GFP. Each image is a 64x64 bounding box surrounding a single cell (center-padded or center-cropped from the original bounding box as necessary). Image intensities are normalized to a minimum of $-1$ and a maximum of 1. Using the set of perturbative perturbations computed in [51] we randomly select 30% to holdout during training for evaluation.

#### B.6.2 GDE modelling architecture

For the encoder architecture, we extend our GNN approach to 2D convolutional layers, standard for image processing. For the generator we use a U-net architecture standard in diffusion for images, but upscaled in expressivity relative to our MNIST and Fashion-MNIST examples.

#### B.6.3 Perturbation Prediction

We find that empirically, our diffusion approach struggles to model the padded border of the cells. So, at inference time we condition on the border to generate our predictions. Using GenePT, we train a ridge regression with grid search (similar to App. B.5) to predict the perturbation distribution embeddings. We also construct a nearest neighbor model using the GenePT embeddings to sample the padding. We then condition on the padding and the predicted latent to sample a set of 1,000 cells from each heldout perturbation. We then compute the DAPI intensity and compare with the ground truth, computing the $R^2$ and the MSE.

### B.7 Methylation atlas of human tissues

#### B.7.1 Simulating raw bisulfite-sequencing reads from methylation patterns

While sample-specific methylation patterns are published in [5], the raw sequencing reads are not public due to patient privacy considerations. Here, we instead use the published methylation patterns (in the form of `.pat` files) to simulate bisulfite sequencing reads. For each methylation site entry of the `.pat` file, we use `wgbstools`[66] to find the 100 preceding bases of the HG38 genome reference, and append to the CpG sequence. We omit all CpG sites with unknown methylation status. We subsample $10^7$ sequencing reads per sample.

#### B.7.2 GDE modelling architecture

We use a 1D convolutional neural network as our encoder, with mean pooling at each layer (analogous to the fully connected GNN with an MLP, but using convolutional layers). For the generator, we use HyenaDNA [35]. We additionally include a linear classification head on top of the distribution embedding, co-trained with a cross-entropy loss.

### B.8 GPRA

#### B.8.1 Data processing details

We collect all sequences in the Gal and Gly conditions from [52] and process them into 100 quantiles by measured expression, totaling 34 million sequences. We one-hot encode these sequences for ACTGN, and tokenize them using the HyenaDNA tokenizer. We break these sequences into 100 quantiles and hold out the top 5 quantiles during training. During training, we construct sets by selecting a "center" quantile and then randomly sampling from that quantile and the two adjacent quantiles.

#### B.8.2 GDE modelling architecture

We use the same architecture as in the methylation experiment (App. B.7).

#### B.8.3 Details for Fig. 8

We encode a random subsample of 130K sequences from each quantile in the Gal condition to construct the set embeddings (the larger dots). We then compute the PCA of these embeddings. We embed all the DNA sequences as sets of size one and project them to the PCA. For the histograms of the TFBS motifs we leverage the PWMs from [53]. We wrote a simple unidirectional motif scanning procedure in Torch to facilitate efficient scanning, and used a threshold of 5 to determine hits. We then sum over the motifs to derive the motif count per sequence, and then compute the histogram by plotting the distribution of these counts by quantile.

### B.9 Spatiotemporal distribution of viral lineages

#### B.9.1 Data preprocessing details

We obtain all SARS-CoV2 spike sequences deposited up to April 2025 in GISAID [58]. We group sequences by submission month and lab of collection. We discard sequences with improperly formatted date fields. During tokenization, we truncate sequences to 1000 amino acids.

#### B.9.2 GDE modelling architecture

The encoder couples the ESM-50M [56] architecture coupled to a mean-pooled GNN, while the generator uses the Progen2-150M architecture [34] with prefix conditioning. We initialize (but do not freeze) the protein language models with their pretrained weights. We use a 128 dimensional latent space.

## C    Background

### C.1    Frequentist, Bayesian, and Predictive Sufficiency

Sufficiency is a classical notion in statistics that formalizes when a statistic retains all information about a parameter or distribution. In this appendix, we distinguish three forms of sufficiency relevant to modern generative modeling and provide canonical examples.

#### C.1.1    Frequentist Sufficiency

Let $\{P_\theta : \theta \in \Theta\}$ be a parametric family of probability distributions on a sample space $\mathcal{X}$. A statistic $T(X_1, \ldots, X_n)$ is *frequentist sufficient* for $\theta$ if the conditional distribution of the data given $T$ does not depend on $\theta$:

$$P_\theta(X_1, \ldots, X_n \mid T(X_1, \ldots, X_n)) = \text{(independent of } \theta).$$

Intuitively, the likelihood depends on the data only through $T$.

#### C.1.2    Bayesian Sufficiency

Given a prior $\pi(\theta)$ over the parameter space, a statistic $T$ is *Bayesian sufficient* for $\theta$ if the posterior depends on the data only through $T$:

$$\pi(\theta \mid X_1, \ldots, X_n) = \pi(\theta \mid T(X_1, \ldots, X_n)).$$

Bayesian sufficiency holds if and only if $T$ is a sufficient statistic in the sense that the posterior is conditionally independent of the data given $T$.

#### C.1.3    Predictive Sufficiency

A weaker notion, often relevant in nonparametric and distributional settings, is *predictive sufficiency*. Assume a joint model for $(\theta, X_{1:n}, X_{\text{new}})$ (e.g. $\theta \sim \pi$ and $X_i \mid \theta \overset{\text{i.i.d.}}{\sim} P_\theta$). A statistic $T$ is predictive sufficient if the distribution of a new sample $X_{\text{new}}$ given $T$ is the same as given the full data:

$$\mathbb{P}(X_{\text{new}} \in B \mid T(X_1, \ldots, X_n)) = \mathbb{P}(X_{\text{new}} \in B \mid X_1, \ldots, X_n), \quad \forall B \in \mathcal{B}(\mathcal{X}).$$

This requires only that $T$ contains enough information to match the predictive distribution of future data.

#### C.1.4    Implications and Comparisons

There is a strict hierarchy among these definitions:

$$\text{Frequentist sufficiency} \Rightarrow \text{Bayesian sufficiency} \Rightarrow \text{Predictive sufficiency}.$$

The first implication follows from the factorization of the likelihood, and the second follows because the posterior predictive is a marginal of the posterior. However, the reverse implications do not hold in general, especially in infinite-dimensional or nonparametric models. In particular, predictive sufficiency may hold in settings where no finite-dimensional parameter exists.

#### C.1.5    Examples

**Example 1** (Gaussian Mean)**.** Let $X_1, \ldots, X_n \sim \mathcal{N}(\mu, \sigma^2)$ with known $\sigma^2$. Then the sample mean $\bar{X}_n$ is sufficient for $\mu$ in all three senses: frequentist, Bayesian, and predictive. The likelihood, posterior, and predictive distributions all depend on the data only through $\bar{X}_n$.

**Example 2** (Uniform$(0, \theta)$)**.** Let $X_1, \ldots, X_n \sim \text{Unif}(0, \theta)$. Then the sample maximum

$$T_n = \max\{X_1, \ldots, X_n\}$$

is the minimal sufficient statistic for $\theta$ in both the frequentist and Bayesian senses. It also suffices for prediction of future samples, since the predictive distribution under $\theta$ is uniform on $[0, \theta]$, and $T_n$ provides all information about $\theta$.

#### C.1.6    Nonparametric Extensions

In the nonparametric regime where $P$ is not indexed by a finite-dimensional parameter, predictive sufficiency remains well-defined. For instance, under a de Finetti (exchangeable) model with a latent random measure $P \sim \Pi$ and $X_i \mid P \overset{\text{i.i.d.}}{\sim} P$, the empirical measure $P_n = \frac{1}{n} \sum_{i=1}^n \delta_{X_i}$ (equivalently, the multiset of observations) is Bayesian and hence predictive sufficient for $P$. In this setting, stronger finite-dimensional forms of sufficiency may not exist, but predictive sufficiency still supports meaningful generative modeling.

## C.2 Otto's Geometry and Statistical Submanifolds

This appendix recalls Otto's Riemannian calculus on the 2-Wasserstein space $\mathcal{P}_2(\mathcal{X})$ and explains how a finite-dimensional parametric family of measures inherits an induced geometry [3]. Throughout, statements are intended in the standard "Otto calculus" sense; rigorous treatments interpret $\mathcal{P}_2$ as a geodesic metric space and identify tangent objects for absolutely continuous measures.

### C.2.1 Wasserstein Space and the Benamou–Brenier Formulation

Let $\mathcal{X} \subseteq \mathbb{R}^d$ be convex (e.g. $\mathcal{X} = \mathbb{R}^d$), and let $\mathcal{P}_2(\mathcal{X})$ be the Borel probability measures on $\mathcal{X}$ with finite second moment. The 2-Wasserstein distance is

$$W_2^2(\mu_0, \mu_1) := \inf_{\gamma \in \Pi(\mu_0, \mu_1)} \int_{\mathcal{X} \times \mathcal{X}} \|x - y\|^2 \, d\gamma(x, y).$$

Benamou–Brenier gives the dynamic formulation

$$W_2^2(\mu_0, \mu_1) = \inf_{\substack{(\mu_t, v_t) \\ \partial_t \mu_t + \nabla \cdot (\mu_t v_t) = 0}} \int_0^1 \int_{\mathcal{X}} \|v_t(x)\|^2 \, d\mu_t(x) \, dt,$$

where the continuity equation holds in the distributional sense, and $v_t \in L^2(\mu_t)$.

### C.2.2 Otto's Riemannian Structure

For $\mu$ absolutely continuous with density $\rho$, a tangent vector can be represented as

$$\dot{\mu} = -\nabla \cdot (\rho \nabla \phi),$$

for a potential $\phi$ (defined up to an additive constant). Equivalently, one represents the tangent direction by its *minimal kinetic energy* velocity field $v = \nabla \phi$. The Otto (Wasserstein) inner product is

$$g_\mu(\dot{\mu}_1, \dot{\mu}_2) = \int_{\mathcal{X}} \nabla \phi_1(x) \cdot \nabla \phi_2(x) \, d\mu(x) = \int_{\mathcal{X}} v_1(x) \cdot v_2(x) \, d\mu(x),$$

with $v_i = \nabla \phi_i$ the minimal-norm representatives. With this metric, constant-speed $W_2$-geodesics are precisely curves of minimal kinetic energy.

If $\mu_0$ is absolutely continuous, the (Brenier) optimal map $T$ from $\mu_0$ to $\mu_1$ induces the displacement interpolation

$$\mu_t = \big((1 - t)\mathrm{id} + tT\big)_{\#} \mu_0,$$

which is a constant-speed $W_2$-geodesic (on convex $\mathcal{X}$).

### C.2.3 Statistical Submanifolds and Induced Wasserstein Geometry

Let $Q$ be a distribution over $\mathcal{P}_2(\mathcal{X})$. To speak of a *submanifold*, we assume $Q$ is supported on a finite-dimensional smooth embedded family

$$\mathcal{M} = \{\mu_\theta : \theta \in \Theta \subset \mathbb{R}^m\} \subset \mathcal{P}_2(\mathcal{X}),$$

where $\theta \mapsto \mu_\theta$ is smooth and $\mu_\theta$ are absolutely continuous.

The induced (pullback) Wasserstein metric on parameters is defined by

$$G_{ij}(\theta) := g_{\mu_\theta}(\partial_i \mu_\theta, \partial_j \mu_\theta) = \int_{\mathcal{X}} \nabla \phi_i \cdot \nabla \phi_j \, d\mu_\theta,$$

where $\phi_i$ solves the elliptic equation

$$-\nabla \cdot (\mu_\theta \nabla \phi_i) = \partial_i \mu_\theta \quad \text{(in distributional sense)}.$$

The intrinsic Riemannian distance on $\mathcal{M}$ can then be written as

$$d_{\mathcal{M}}(\mu_{\theta_0}, \mu_{\theta_1}) = \inf_{\theta_t} \int_0^1 \sqrt{\dot{\theta}_t^\top G(\theta_t) \dot{\theta}_t} \, dt,$$

and satisfies $d_{\mathcal{M}}(\mu_0, \mu_1) \geq W_2(\mu_0, \mu_1)$ in general (strict unless $\mathcal{M}$ is geodesically closed in $\mathcal{P}_2$).

### C.2.4 Examples and Application to GDEs

Examples include Gaussian families (closed under $W_2$-geodesics) and general smooth parametric families. For mixture models with finitely many components, one can study the induced Wasserstein metric on parameters, although ambient $W_2$-geodesics between mixtures typically leave the class.

In this work, we interpret GDEs as learning smooth embeddings of such a constrained family of data-generating distributions into Euclidean latent space; empirically the learned latent geometry may approximate the intrinsic geometry induced by the Wasserstein metric.

# D  Theory

Throughout, let $(\mathcal{X}, d)$ be a Polish metric space and let $\mathcal{B}$ denote its Borel $\sigma$-algebra. Let $P \in \mathcal{P}(\mathcal{X})$ denote a probability law on $\mathcal{X}$. Given $m \in \mathbb{N}$, let $S_m = (X_1, \ldots, X_m)$ be an i.i.d. sample from $P$, and let $P_m = \frac{1}{m} \sum_{i=1}^{m} \delta_{X_i}$ denote the empirical measure.

Let $\mathcal{M}_0(\mathcal{X})$ denote the vector space of finite signed Borel measures on $\mathcal{X}$ with total mass 0, equipped with the bounded–Lipschitz norm $\| \cdot \|_{\mathrm{BL}}$ defined below.

We use $P_1, P_2$ to denote two (possibly distinct) probability laws on $\mathcal{X}$, and $S_1, S_2$ for independent samples from $P_1, P_2$ respectively.

For signed measures $\nu, \mu$ on $(\mathcal{X}, \mathcal{B})$ define

$$d_{\mathrm{BL}}(\nu, \mu) \ := \ \sup_{\substack{f: \mathcal{X} \to [-1,1] \\ \mathrm{Lip}(f) \leq 1}} \left| \int f \, d(\nu - \mu) \right|.$$

We use $\| \cdot \|_{\mathrm{BL}}$ for the corresponding norm $\|\nu\|_{\mathrm{BL}} := d_{\mathrm{BL}}(\nu, 0)$. Since the class of bounded 1-Lipschitz functions is contained in the class of all bounded measurable functions, we have $d_{\mathrm{BL}}(\nu, \mu) \leq \|\nu - \mu\|_{\mathrm{TV}}$.

## D.1  Necessity of Distributional Invariance

**Motivation**  Our goal is to design encoder architectures that flexibly model unknown data distributions while guaranteeing consistent generation of the underlying law as sample size grows. Since the true distribution $P$ is not known in advance, the encoder must be constructed to generalize across all possible $P$, without leaking spurious information tied to the specific realization or sample size. If the encoder depends on sample-level artifacts—such as ordering, multiplicity, or the raw sample size—it may encode features that a generator can exploit, breaking the guarantee that

$$\mathcal{G}(\mathcal{E}(S_m)) \xrightarrow{d} P \quad \text{as } m \to \infty, \quad S_m \sim P^{\otimes m}.$$

This risk arises even under either permutation or proportional invariance on their own: both permit dependencies that vanish only in expectation and are insufficient to ensure correct extrapolation with increasing $m$. For example, encoders based on unnormalized sum aggregations (e.g., DeepSets) will vary with $m$ even when the empirical distribution is unchanged, leading to divergence at inference time.

To formalize this constraint, we appeal to two classical principles from statistical decision theory and invariance: (i) under i.i.d. sampling, the empirical measure is a sufficient statistic for the (nonparametric) model indexed by the unknown law $P$, so conditioning on it loses no information about $P$; (ii) it is also minimal (and the maximal invariant under permutations), meaning it discards exactly the ancillary degrees of freedom (ordering and other sample-level artifacts) that do not carry information about $P$. In our setting this motivates enforcing that the encoder depends on the sample only through the empirical distribution: any additional channels (e.g. ordering or set-size effects) are unnecessary for identifying $P$ and can be spuriously exploited by a flexible generator, especially when extrapolating across set sizes.

We define distributional invariance:

**Definition 1** (Distributional invariance). A family of encoder maps $(\mathcal{E}_m)_{m \geq 1}$ with $\mathcal{E}_m : \mathcal{X}^m \to \mathcal{Z}$ is *distributionally invariant* if there exists a measurable map $\phi : \mathcal{P}(\mathcal{X}) \to \mathcal{Z}$ such that for every $m$ and every $S_m = (x_1, \ldots, x_m) \in \mathcal{X}^m$ with empirical measure $P_m = \frac{1}{m} \sum_{i=1}^{m} \delta_{x_i}$,

$$\mathcal{E}_m(S_m) = \phi(P_m).$$

In particular, any such family is permutation invariant and consistent across set sizes under proportional duplication, i.e. for every integer $K \geq 1$,

$$\mathcal{E}_m(S_m) = \mathcal{E}_{Km}(\underbrace{S_m, \ldots, S_m}_{K \text{ copies}}).$$

**Empirical measure as a lossless and minimal summary.**  Fix $m \in \mathbb{N}$. Consider the (nonparametric) i.i.d. model $\{P^{\otimes m} : P \in \mathcal{P}(\mathcal{X})\}$ on $\mathcal{X}^m$. We use the standard (Neyman–Fisher) notion of sufficiency:

**Definition 2** (Sufficiency). A statistic $T_m : \mathcal{X}^m \to \mathcal{T}$ is *sufficient* for the family $\{P^{\otimes m} : P \in \mathcal{P}(\mathcal{X})\}$ if the conditional distribution of $X_{1:m} \sim P^{\otimes m}$ given $T_m(X_{1:m})$ does not depend on $P$. Equivalently, for every bounded measurable $f : \mathcal{X}^m \to \mathbb{R}$, there exists a measurable $g_f : \mathcal{T} \to \mathbb{R}$ such that for all $P$,

$$\mathbb{E}_{P^{\otimes m}} \big[ f(X_{1:m}) \mid T_m(X_{1:m}) \big] = g_f \big( T_m(X_{1:m}) \big) \quad \text{a.s.}$$

Let $\mathfrak{S}_m$ be the symmetric group acting on $\mathcal{X}^m$ by $(x_1, \ldots, x_m) \mapsto (x_{\pi(1)}, \ldots, x_{\pi(m)})$. Call $T_m$ *permutation invariant* if $T_m(x) = T_m(x_\pi)$ for all $\pi \in \mathfrak{S}_m$.

**Theorem 1** (Empirical measure is sufficient, minimal, and a maximal invariant). *Let $X_{1:m} = (X_1, \ldots, X_m) \sim P^{\otimes m}$ and $P_m = \frac{1}{m} \sum_{i=1}^m \delta_{X_i}$. Then:*

(i) (**Sufficiency / "losing nothing"**). *$P_m$ is sufficient in the sense of Definition 2. Moreover, for any bounded measurable $f : \mathcal{X}^m \to \mathbb{R}$, a version of the conditional expectation is given by symmetrisation:*

$$\mathbb{E}_{P^{\otimes m}}\big[f(X_{1:m}) \mid P_m\big] = \frac{1}{m!} \sum_{\pi \in \mathfrak{S}_m} f\big(X_{\pi(1)}, \ldots, X_{\pi(m)}\big) \quad a.s.,$$

*which does not depend on $P$. In particular, the sample ordering is ancillary given $P_m$.*

(ii) (**Maximal invariant / all permutation-invariant summaries factor through $P_m$**). *Let $\mathcal{P}_m(\mathcal{X}) := \{m^{-1} \sum_{i=1}^m \delta_{x_i} : x_{1:m} \in \mathcal{X}^m\}$ be the set of empirical measures with $m$ atoms (counting multiplicity). If $T_m : \mathcal{X}^m \to \mathcal{T}$ is permutation invariant, then there exists a measurable $\phi_m : \mathcal{P}_m(\mathcal{X}) \to \mathcal{T}$ such that $T_m = \phi_m \circ P_m$ pointwise on $\mathcal{X}^m$. (One may extend $\phi_m$ to all of $\mathcal{P}(\mathcal{X})$ arbitrarily if desired.)*

(iii) (**Minimal sufficient / "not keeping anything unnecessary"**). *If $T_m$ is sufficient for $\{P^{\otimes m} : P \in \mathcal{P}(\mathcal{X})\}$, then $P_m$ is measurable with respect to $\sigma(T_m)$: there exists a measurable $h_m : \mathcal{T} \to \mathcal{P}(\mathcal{X})$ such that*

$$P_m = h_m\big(T_m(X_{1:m})\big) \quad P^{\otimes m}\text{-a.s. for every } P \in \mathcal{P}(\mathcal{X}).$$

*Equivalently, $\sigma(P_m)$ is the minimal sufficient $\sigma$-field.*

*Proof sketch.* (i) Let $f$ be bounded measurable and define the symmetrisation $\bar{f}(x_{1:m}) := \frac{1}{m!} \sum_{\pi \in \mathfrak{S}_m} f\big(x_{\pi(1)}, \ldots, x_{\pi(m)}\big)$. Then $\bar{f}$ is permutation invariant, hence depends on $x_{1:m}$ only through its multiset, i.e. only through $P_m$. For any bounded measurable $H$ that is $\sigma(P_m)$-measurable, $H(X_{1:m}) = H(X_{\pi(1)}, \ldots, X_{\pi(m)})$ for all $\pi$, so by exchangeability,

$$\mathbb{E}\big[f(X_{1:m})H(X_{1:m})\big] = \mathbb{E}\Big[\frac{1}{m!} \sum_\pi f\big(X_{\pi(1)}, \ldots, X_{\pi(m)}\big) H(X_{1:m})\Big] = \mathbb{E}\big[\bar{f}(X_{1:m})H(X_{1:m})\big].$$

Thus $\bar{f}(X_{1:m})$ is a version of $\mathbb{E}[f(X_{1:m}) \mid P_m]$, and it does not depend on $P$, establishing sufficiency.

(ii) If $x, y \in \mathcal{X}^m$ satisfy $P_m(x) = P_m(y)$, then $y$ is a permutation of $x$, so permutation invariance gives $T_m(x) = T_m(y)$. Hence $T_m$ is constant on the fibres of the measurable map $P_m : \mathcal{X}^m \to \mathcal{P}_m(\mathcal{X})$, i.e. $T_m$ is $\sigma(P_m)$-measurable. By the Doob–Dynkin lemma, there exists a measurable $\phi_m : \mathcal{P}_m(\mathcal{X}) \to \mathcal{T}$ such that $T_m = \phi_m \circ P_m$.

(iii) If $T_m(x) = T_m(y)$ but $P_m(x) \neq P_m(y)$, choose an atomic law $P$ that puts positive mass on every point appearing in $x$ or $y$. Then both sequences have positive probability under $P^{\otimes m}$. By varying the atomic masses on that finite support, the ratio $P^{\otimes m}(\{x\})/P^{\otimes m}(\{y\})$ can be changed while keeping $T_m(x) = T_m(y)$, forcing the conditional distribution of $X_{1:m}$ given $T_m$ to depend on $P$, contradicting sufficiency. Therefore $T_m(x) = T_m(y) \Rightarrow P_m(x) = P_m(y)$, so $P_m$ is measurable with respect to $\sigma(T_m)$. $\qquad\square$

**Corollary 1** (Necessity of empirical-measure dependence for artifact-free encoders). Let $Z_m = \mathcal{E}_m(X_{1:m})$ be an encoder output.

(i) If $\mathcal{E}_m$ is permutation invariant, then by Theorem 1(ii) there exists $\phi_m$ such that $Z_m = \phi_m(P_m)$: the encoder can only depend on the data through the empirical distribution.

(ii) If, additionally, the family $(\mathcal{E}_m)_{m \geq 1}$ is distributionally invariant in the sense of Definition 1, then there exists a single measurable map $\phi : \mathcal{P}(\mathcal{X}) \to \mathcal{Z}$ such that $Z_m = \phi(P_m)$ for all $m$.

In this sense, restricting to empirical-measure dependence discards only ancillary sample-level degrees of freedom (order and size artifacts) and keeps exactly the information relevant to the underlying law $P$.

**Remark** (Connection to predictive sufficiency and scaling). *Theorem 1 formalises "losing nothing" (sufficiency) and "not keeping anything unnecessary" (minimality) for the nonparametric i.i.d. model. Definition 7 in Appendix D.3 is an asymptotic, reconstruction-based analogue restricted to the manifold $\mathcal{M}$: it asks that from a low-dimensional coordinate $\phi(P_m)$ one can reconstruct $P_m$ (in $\|\cdot\|_{\mathrm{BL}}$) as $m \to \infty$. Operationally, enforcing duplication invariance removes a particularly dangerous ancillary channel: set size. Without it (e.g. sum pooling), a flexible generator can fit finite-$m$ size effects during training and behave unpredictably when $m$ changes, even if the empirical distribution is unchanged.*

## D.2  A Complete Large-$m$ Analysis of the Plug-in Loss

**Motivation**  We analyze the statistical properties of the plug-in loss used to train distributional encoders and generators. Our goal is to understand the asymptotic behavior of this loss as the sample size grows, and to establish conditions under which the learned generator recovers the true data distribution. This analysis provides a principled foundation for the training objectives used in our framework.

**Setting**   First we establish some notation and definitions.

**Definition 3** (Hadamard differentiability). A map $T : \mathcal{D} \to \mathcal{Y}$ between normed spaces is *Hadamard differentiable* at $x \in \mathcal{D}$ if there exists a continuous linear operator $DT_x$ such that for every sequence $h_t \to h$ in $\mathcal{D}$ and $t \downarrow 0$, $\frac{T(x+th_t)-T(x)}{t} \longrightarrow DT_x[h]$.

**Definition 4** (Fréchet differentiability). Let $T : \mathcal{D} \to \mathcal{Y}$ be a map between normed vector spaces. We say that $T$ is *Fréchet differentiable* at $x \in \mathcal{D}$ if there exists a bounded linear operator $A : \mathcal{D} \to \mathcal{Y}$ such that

$$\lim_{\|h\|_{\mathcal{D}} \to 0} \frac{\|T(x+h) - T(x) - A(h)\|_{\mathcal{Y}}}{\|h\|_{\mathcal{D}}} = 0.$$

The operator $A$ is called the Fréchet derivative of $T$ at $x$.

**Definition 5** (Tangent set at $Q_0$.). Let $Q_0 \in \mathcal{P}(\mathcal{X})$ and write $\mathcal{M}_0(\mathcal{X})$ for finite signed Borel measures on $\mathcal{X}$ with total mass 0. Define the $L^2(Q_0)$–tangent space

$$\mathbb{D}_0(Q_0) := \left\{ h \in \mathcal{M}_0(\mathcal{X}) \ : \ h \ll Q_0, \ \frac{dh}{dQ_0} \in L^2(Q_0) \right\}, \qquad \|h\|_{\mathbb{D}_0} := \left\| \frac{dh}{dQ_0} \right\|_{L^2(Q_0)}.$$

We view $\mathbb{D}_0(Q_0)$ as a normed linear space via the identification $h \leftrightarrow dh/dQ_0$.

We work in the following general setting:

**Assumption 1** (Data and Empirical Measure). *$(\mathcal{X}, \mathcal{B})$ is a Polish space; $P \in \mathcal{P}(\mathcal{X})$ is the true data law. Observations $S_m = (X_1, \ldots, X_m)$ are i.i.d. $P$. The empirical measure is $P_m = \frac{1}{m} \sum_{i=1}^{m} \delta_{X_i}$.*

**Assumption 2** (Encoder regularity). *For each probability law $P \in \mathcal{P}(\mathcal{X})$ the encoder $\phi : \mathcal{P}(\mathcal{X}) \to \mathbb{R}^d$ satisfies*

(i) ***Distributional invariance:*** *$\mathcal{E}_m(S_m) = \phi(P_m)$ depends on the sample only via its empirical measure.*

(ii) ***Pathwise (Hadamard) differentiability:*** *$\phi$ is pathwise differentiable at $P$ and its canonical gradient[1] $\psi_P : \mathcal{X} \to \mathbb{R}^d$ belongs to $L^2(P)$.*

(iii) ***Asymptotic linearity (AL):*** *there exists a remainder $r_m$ such that*

$$\sqrt{m} \left\{ \phi(P_m) - \phi(P) \right\} = \frac{1}{\sqrt{m}} \sum_{i=1}^{m} \psi_P(X_i) + r_m,$$

*where $\mathbb{E}_{X \sim P}[\psi_P(X)] = 0$ and $r_m \to 0$ in $L^2(P^{\otimes m})$.*

*In particular,*

$$\sqrt{m} \left\{ \phi(P_m) - \phi(P) \right\} \ \overset{d}{\Rightarrow} \ \mathcal{N}(0, \Sigma_\phi), \quad \Sigma_\phi := \mathrm{Var}_{X \sim P}[\psi_P(X)],$$

*and $\sup_m \mathbb{E} \left\| \sqrt{m} \left\{ \phi(P_m) - \phi(P) \right\} \right\|^2 < \infty$.*

**Assumption 3** (Generator). *Let $\mathcal{M}(\mathcal{X})$ denote the vector space of finite signed Borel measures on $\mathcal{X}$ equipped with $\| \cdot \|_{\mathrm{BL}}$, and identify $\mathcal{P}(\mathcal{X}) \subset \mathcal{M}(\mathcal{X})$. Assume the generator $\mathcal{G} : \mathbb{R}^d \to \mathcal{P}(\mathcal{X})$ admits a local Fréchet expansion at $\mu := \phi(P)$ when viewed as a map into $\mathcal{M}(\mathcal{X})$: there exists a bounded linear map $D_\mu \mathcal{G} : \mathbb{R}^d \to \mathcal{M}_0(\mathcal{X})$ such that*

$$\left\| \mathcal{G}(\mu + h) - \mathcal{G}(\mu) - D_\mu \mathcal{G}[h] \right\|_{\mathrm{BL}} = o(\|h\|_{\mathbb{R}^d}) \qquad \text{as } \|h\| \to 0.$$

*Moreover, writing $Q_0 := \mathcal{G}(\mu)$, the derivative is $L^2(Q_0)$–compatible:*

$$D_\mu \mathcal{G}(\mathbb{R}^d) \subseteq \mathbb{D}_0(Q_0) \quad \text{and} \quad \sup_{\|h\| \leq 1} \left\| D_\mu \mathcal{G}[h] \right\|_{\mathbb{D}_0} < \infty.$$

*Finally, the remainder is negligible in the $L^2(Q_0)$ tangent norm:*

$$\left\| \mathcal{G}(\mu + h) - \mathcal{G}(\mu) - D_\mu \mathcal{G}[h] \right\|_{\mathbb{D}_0(Q_0)} = o(\|h\|_{\mathbb{R}^d}) \qquad \text{as } \|h\| \to 0.$$

**Assumption 4** (Divergence (Hadamard differentiability on an $L^2$ tangent space)). *Let $Q_0 := \mathcal{G}(\mu)$ and $\mathbb{D}_0(Q_0)$ be as defined above. The discrepancy $\mathcal{L} : \mathcal{P}(\mathcal{X})^2 \to \mathbb{R}_+$ satisfies:*

(i) ***(Hadamard differentiability on $\mathbb{D}_0(Q_0)$)*** *for each fixed $P$, the map $Q \mapsto \mathcal{L}(P, Q)$ is Hadamard differentiable at $Q_0$ tangentially to $\mathbb{D}_0(Q_0)$ (equipped with $\| \cdot \|_{\mathbb{D}_0}$), with continuous linear derivative*

$$D_2 \mathcal{L}(P, Q_0) : \mathbb{D}_0(Q_0) \to \mathbb{R}.$$

---

[1] In the semiparametric sense of 67, i.e. the unique influence function representing the functional derivative along $\mathcal{M}_0(\mathcal{X})$.

*(ii) (**Separating property**) $\mathcal{L}(P,Q) = 0 \implies P = Q$;*

*(iii) (**Weak-continuity**) for each fixed $P$, if $\mathcal{L}(P,Q_n) \to 0$ then $Q_n \Rightarrow P$.*

We work with the discrepancy $\mathcal{L}$ from Assumption 4.

The *plug-in loss* is

$$\widehat{\ell}_m := \mathcal{L}\big(P,\, \mathcal{G}(\phi(P_m))\big)$$

and the *population loss* is

$$\ell^* := \mathcal{L}\big(P,\, \mathcal{G}(\phi(P))\big),$$

where $P_m$ is the empirical measure of the sample, $\phi$ is the encoder, and $\mathcal{G}$ is the generator. When $P$ is unknown, $\mathcal{L}(P,\cdot)$ is evaluated via an empirical Monte Carlo estimate based on $S_m$; the results below describe the additional error incurred by using $\phi(P_m)$ in place of $\phi(P)$.

**Lemma 1** (Functional Delta Method, [67, Thm. 3.9.4]). *Let $(\mathbb{D}, \|\cdot\|_{\mathbb{D}})$ and $(\mathbb{E}, \|\cdot\|_{\mathbb{E}})$ be normed vector spaces. Let $T : \mathbb{D} \to \mathbb{E}$ be a map that is Hadamard differentiable at a point $z \in \mathbb{D}$ tangentially to a subset $\mathbb{D}_0 \subseteq \mathbb{D}$, with continuous linear derivative denoted $DT_z : \mathbb{D}_0 \to \mathbb{E}$.*

*Suppose:*

*(a) There exist random elements $Z_m$ taking values in $\mathbb{D}$ such that:*

$$\sqrt{m}(Z_m - z) \xrightarrow{d} Z$$

*for some tight limit $Z$ taking values in $\mathbb{D}_0$.*

*(b) $Z$ is tight and Borel measurable.*

*Then:*

$$\sqrt{m}\big(T(Z_m) - T(z)\big) \xrightarrow{d} DT_z(Z),$$

*where $DT_z(Z)$ is a random element of $\mathbb{E}$.*

*In particular, if $Z$ is Gaussian in $\mathbb{D}_0$ and $DT_z$ is continuous and linear, then $DT_z(Z)$ is Gaussian in $\mathbb{E}$.*

## Main Result

**Theorem 2** (Large-$m$ behaviour of the plug-in loss). *Assume 1, 2, 3, and 4.*

*Let $\mu := \phi(P)$ and $\widehat{\ell}_m := \mathcal{L}\big(P,\, \mathcal{G}(\phi(P_m))\big)$. Then:*

*(a) **Asymptotic normality of the Encoder.***

$$\sqrt{m}\big\{\phi(P_m) - \phi(P)\big\} \xRightarrow{d} \mathcal{N}\big(0, \Sigma_\phi\big), \qquad \Sigma_\phi := \mathrm{Var}_{X \sim P}\big[\psi_P(X)\big].$$

*(b) **Consistency (and mean consistency) of the loss.** Writing $\ell(\theta) := \mathcal{L}\big(P, \mathcal{G}(\theta)\big)$, we have*

$$\widehat{\ell}_m = \ell\big(\phi(P_m)\big) \xrightarrow[m\to\infty]{\mathbb{P}} \ell(\mu) =: \ell^*.$$

*If, in addition, $\ell$ is locally Lipschitz on a neighbourhood of $\mu$ (as a map $\mathbb{R}^d \to \mathbb{R}$) and the sequence $\{\ell(\phi(P_m))\}_{m\geq 1}$ is uniformly integrable, then*

$$\mathbb{E}\big|\widehat{\ell}_m - \ell^*\big| \to 0 \quad \text{and hence} \quad \mathbb{E}[\widehat{\ell}_m] \to \ell^*.$$

*A sufficient condition for uniform integrability is the following growth bound: there exist $p > 1$ and $C < \infty$ such that $|\ell(\theta)| \leq C\,(1 + \|\theta\|^p)$ for all $\theta$ and $\sup_m \mathbb{E}\|\phi(P_m)\|^p < \infty$.*

*Moreover, if $\ell$ is twice continuously differentiable in a neighbourhood of $\mu$ with bounded Hessian, then a second-order Taylor expansion yields*

$$\mathbb{E}[\widehat{\ell}_m] - \ell^* = D\ell_\mu\big[\mathbb{E}[\phi(P_m) - \mu]\big] + O(m^{-1}) = o(m^{-1/2}).$$

*In common unbiased cases where $\mathbb{E}[\phi(P_m) - \mu] = O(m^{-1})$ (e.g. sample means and many smooth $M$-estimators), this simplifies to $\mathbb{E}[\widehat{\ell}_m] - \ell^* = O(m^{-1})$.*

*(c) **Asymptotic normality of the loss.** Let $\ell(\theta) := \mathcal{L}\big(P, \mathcal{G}(\theta)\big)$ and denote its derivative at $\mu$ by the continuous linear functional*

$$D\ell_\mu : \mathbb{R}^d \to \mathbb{R}, \qquad D\ell_\mu[h] := D_2\mathcal{L}\big(P, Q_0\big)\big[\,D_\mu\mathcal{G}[h]\,\big], \quad Q_0 := \mathcal{G}(\mu).$$

*Then*

$$\sqrt{m}\,(\widehat{\ell}_m - \ell^*) \xRightarrow{d} \mathcal{N}\big(0, \sigma^2\big), \qquad \sigma^2 = D\ell_\mu\, \Sigma_\phi\, D\ell_\mu^\top.$$

*(Identifying $D\ell_\mu$ with a gradient vector $\nabla_\mu \ell \in \mathbb{R}^d$ under the Euclidean inner product yields $\sigma^2 = (\nabla_\mu \ell)^\top \Sigma_\phi (\nabla_\mu \ell)$.)*

*(d)* ***Consistency under correct specification.*** *Fix P. If $(\phi^\star, \mathcal{G}^\star)$ minimizes the population objective*

$$(\phi, \mathcal{G}) \longmapsto \mathcal{L}\big(P, \mathcal{G}(\phi(P))\big)$$

*over the model class, and if the model is well-specified in the sense that the minimum value is $0$ (equivalently, $\mathcal{L}(P, \mathcal{G}^\star(\phi^\star(P))) = 0$), then $\mathcal{G}^\star(\phi^\star(P_m)) \Rightarrow P$ in $P^{\otimes m}$-probability as $m \to \infty$.*

*Proof.* **Step 1: Asymptotic Normality of the encoder (a).** Assumption 2(iii) (asymptotic linearity) gives

$$\sqrt{m}\big\{\phi(P_m) - \phi(P)\big\} = \frac{1}{\sqrt{m}} \sum_{i=1}^{m} \psi_P(X_i) + o_p(1),$$

and the classical multivariate CLT yields the stated convergence.

Let $\Delta_m := \phi(P_m) - \mu$ so that, by (a), $\sqrt{m}\,\Delta_m \overset{d}{\Rightarrow} \mathcal{N}(0, \Sigma_\phi)$.

**Step 2 (consistency and mean consistency).** Write $\ell(\theta) := \mathcal{L}\big(P, \mathcal{G}(\theta)\big)$ and $\Delta_m := \phi(P_m) - \mu$. By Assumption 2(iii), $\Delta_m = O_{\mathbb{P}}(m^{-1/2})$, hence $\phi(P_m) = \mu + \Delta_m \to \mu$ in probability. Since $\ell$ is continuous at $\mu$ (automatic if $\ell$ is differentiable at $\mu$), the continuous mapping theorem gives $\widehat{\ell}_m = \ell(\mu + \Delta_m) \to \ell(\mu) = \ell^*$ in probability.

For convergence of expectations, assume $\ell$ is locally Lipschitz near $\mu$ and $\{\ell(\phi(P_m))\}_{m \geq 1}$ is uniformly integrable. Since $\phi(P_m) \to \mu$ in probability and $\ell$ is continuous at $\mu$, we have $\ell(\phi(P_m)) \to \ell(\mu)$ in probability, i.e. $\widehat{\ell}_m \to \ell^*$ in probability. Uniform integrability then implies $\mathbb{E}[\widehat{\ell}_m] \to \ell^*$ and $\mathbb{E}|\widehat{\ell}_m - \ell^*| \to 0$.

If one prefers a direct bound using local Lipschitz, let $U$ be a neighbourhood of $\mu$ on which $|\ell(\theta) - \ell(\mu)| \leq L\|\theta - \mu\|$. Then

$$\mathbb{E}|\widehat{\ell}_m - \ell^*| \leq \mathbb{E}[|\ell(\mu + \Delta_m) - \ell(\mu)|\, \mathbf{1}\{\mu + \Delta_m \in U\}] + \mathbb{E}[|\ell(\mu + \Delta_m) - \ell(\mu)|\, \mathbf{1}\{\mu + \Delta_m \notin U\}].$$

The first term is $\leq L\,\mathbb{E}\|\Delta_m\|$ and tends to $0$ since $\sup_m \mathbb{E}\|\sqrt{m}\,\Delta_m\|^2 < \infty$. The second term vanishes by uniform integrability together with $\mathbb{P}(\mu + \Delta_m \notin U) \to 0$.

If $\ell$ is $C^2$ near $\mu$ with bounded Hessian, a second-order Taylor expansion gives

$$\mathbb{E}[\widehat{\ell}_m] - \ell^* = D\ell_\mu\big[\mathbb{E}[\Delta_m]\big] + O\big(\mathbb{E}\|\Delta_m\|^2\big) = D\ell_\mu\big[\mathbb{E}[\Delta_m]\big] + O(m^{-1}) = o(m^{-1/2}),$$

since $\mathbb{E}[\Delta_m] = \mathbb{E}[r_m]/\sqrt{m} = o(m^{-1/2})$ under Assumption 2(iii). (If additionally $\mathbb{E}[\Delta_m] = O(m^{-1})$, then the bias is $O(m^{-1})$.)

**Step 3: Asymptotic Normality of the loss (c).** Let $\Delta_m := \phi(P_m) - \mu$ so that $\sqrt{m}\,\Delta_m \Rightarrow Z$ with $Z \sim \mathcal{N}(0, \Sigma_\phi)$ by part (a).

By Assumption 3 (including the $\mathbb{D}_0(Q_0)$ remainder control),

$$\sqrt{m}\big\{\mathcal{G}(\mu + \Delta_m) - Q_0\big\} = \sqrt{m}\, D_\mu \mathcal{G}[\Delta_m] + o_p(1) \quad \text{in } \big(\mathbb{D}_0(Q_0), \|\cdot\|_{\mathbb{D}_0}\big).$$

Since $D_\mu \mathcal{G} : \mathbb{R}^d \to \mathbb{D}_0(Q_0)$ is bounded linear, we also have $\sqrt{m}\, D_\mu \mathcal{G}[\Delta_m] \Rightarrow D_\mu \mathcal{G}[Z]$ in $\mathbb{D}_0(Q_0)$.

Now apply the functional delta method (Lemma 1) to the map $Q \mapsto \mathcal{L}(P, Q)$ at $Q_0$, tangentially to $\mathbb{D}_0(Q_0)$:

$$\sqrt{m}\big\{\mathcal{L}\big(P, \mathcal{G}(\mu + \Delta_m)\big) - \mathcal{L}(P, Q_0)\big\} \Rightarrow D_2\mathcal{L}(P, Q_0)\big[D_\mu \mathcal{G}[Z]\big].$$

Define the continuous linear functional $D\ell_\mu : \mathbb{R}^d \to \mathbb{R}$ by

$$D\ell_\mu[h] := D_2\mathcal{L}(P, Q_0)\big[D_\mu \mathcal{G}[h]\big].$$

Then the limit is $D\ell_\mu[Z]$, which is Gaussian with variance $\sigma^2 = D\ell_\mu\, \Sigma_\phi\, D\ell_\mu^\top$.

**Step 4: Consistency under correct specification (d).** If $(\phi^\star, \mathcal{G}^\star)$ minimises $P \mapsto \mathcal{L}\big(P, \mathcal{G}(\phi(P))\big)$ and the model is well specified, then $\mathcal{L}\big(P, \mathcal{G}^\star(\phi^\star(P))\big) = 0$, so $\mathcal{G}^\star(\phi^\star(P)) = P$ by Assumption 4(ii). Repeating the expansion from (c) with $(\phi^\star, \mathcal{G}^\star)$ shows that

$$\mathcal{L}\big(P, \mathcal{G}^\star(\phi^\star(P_m))\big) = O_{\mathbb{P}}(m^{-1/2}),$$

hence $\mathcal{L}\big(P, \mathcal{G}^\star(\phi^\star(P_m))\big) \to 0$ in probability. Finally, Assumption 4(iii) implies that the topology induced by $\mathcal{L}(P, \cdot)$ is at least as strong as the weak topology: for every $\eta > 0$ there exists $\delta > 0$ such that $\mathcal{L}(P, Q) < \delta$ entails $d_{\mathrm{BL}}(P, Q) < \eta$. (Otherwise one could construct a sequence $(Q_n)$ with $\mathcal{L}(P, Q_n) \to 0$ but $d_{\mathrm{BL}}(P, Q_n) \geq \eta$ for all $n$, contradicting the assumption.)

Since $\mathcal{L}\big(P, \mathcal{G}^\star(\phi^\star(P_m))\big) \to 0$ in probability, for any fixed $\eta > 0$ we may choose $\delta > 0$ as above and obtain

$$\mathbb{P}\Big(d_{\mathrm{BL}}\big(\mathcal{G}^\star(\phi^\star(P_m)), P\big) > \eta\Big) \ \leq\ \mathbb{P}\Big(\mathcal{L}\big(P, \mathcal{G}^\star(\phi^\star(P_m))\big) \geq \delta\Big) \xrightarrow[m\to\infty]{} 0.$$

Thus $\mathcal{G}^\star(\phi^\star(P_m)) \Rightarrow P$ in probability as claimed.

$\square$

**Encoders: examples, counter-examples, and CLTs**    The only encoder requirement entering Theorem 2 is Assumption 2. We now show that it is satisfied by a large family of permutation-invariant architectures built from *asymptotically-linear $(M/Z)$ poolers*.

**Generic $K$-layer pool–concat encoder**    Fix $K \in \mathbb{N}$. Given a set of samples $S_m = \{x_1, \dots, x_m\}$ define recursively

$$h_i^{(0)} \ = \ \rho(x_i), \quad \bar{h}^{(\ell)} \ = \ T^{(\ell)}\big(h_{1:m}^{(\ell-1)}\big), \quad h_i^{(\ell)} \ = \ \mathrm{MLP}_\ell\big(h_i^{(\ell-1)}, \bar{h}^{(\ell)}\big), \qquad \ell = 1, \dots, K,$$

and set the encoder output to be another pooler $\phi(P_m) = T^{(K+1)}\big(h_{1:m}^{(K)}\big)$.

We call a permutation-invariant functional an *asymptotically linear (AL) pooler* if it is root-$m$ consistent and admits an influence-function expansion; precise details follow.

**Definition 6** (Asymptotically-linear pooler)**.**  Let $\varphi : \mathcal{P}(\mathcal{X}) \to \mathbb{R}^d$ be a fixed statistical functional. A *family of symmetric maps* $(T_m)_{m \geq 1}$ with $T_m : \mathcal{X}^m \to \mathbb{R}^d$ is an *AL pooler* for $\varphi$ at law $P$ if each $T_m$ depends on the sample only through its empirical measure $P_m$ and there exists $\psi_P \in L^2(P)$ such that, as $m \to \infty$,

$$\sqrt{m}\,\big\{T_m(X_{1:m}) - \varphi(P)\big\} \ = \ \frac{1}{\sqrt{m}} \sum_{i=1}^m \psi_P(X_i) \ + \ o_p(1).$$

Examples include the sample mean, median, trimmed mean, Huber $M$-estimators, M-quantiles, and studentised $Z$-estimators with finite variance.

**Proposition 4** (CLT for $K$-layer AL pool–concat encoders)**.**  *Assume*

(i) *each $T^{(\ell)}$ ($\ell = 1, \dots, K+1$) is a distributionally invariant AL pooler (in the sense of Definition 6) at $P$;*

(ii) *each $\mathrm{MLP}_\ell$ and the base feature map $\rho : \mathcal{X} \to \mathbb{R}^p$ are $C^2$ with bounded derivatives, and weights are frozen as $m \to \infty$.*

*Then the encoder $\phi$ is distributionally invariant, pathwise differentiable, and satisfies the CLT of Assumption 2 with*

$$\sqrt{m}\big\{\phi(P_m) - \phi(P)\big\} \ \overset{d}{\Rightarrow} \ \mathcal{N}\big(0, \Sigma_\phi\big),$$

*for some finite covariance matrix $\Sigma_\phi$.*

*Sketch.*  The composition of Lipschitz maps ($\mathrm{MLP}_\ell$) with AL poolers is Hadamard differentiable by repeated application of the delta method (iterating Lemma 1, [67]). Plugging each AL expansion into the chain yields an overall AL expansion whose leading empirical-process term is $m^{-1/2} \sum_{i=1}^m \psi_P^\star(X_i)$ for some $L^2(P)$ function $\psi_P^\star$, giving the CLT.  $\square$

**Instantiation to common architectures**

**Corollary 2** (DeepSets, Transformers without positional enc.)**.**  Encoder architectures of either type below satisfy Assumption 2 and Proposition 4:

(a) *DeepSets / fully-connected GNN with global mean:* $T^{(\ell)}$ and $T^{(K+1)}$ are sample means;

(b) *Self-attention block with mean head:* $T^{(\ell)}$ are sample means; $\mathrm{MLP}_\ell$ includes the softmax-attention update.

**Why max-pooling fails**    The max functional $T_{\max}(x_{1:m}) = \max_i x_i$ is *not* Hadamard differentiable at continuous laws: its influence function is identically 0 whenever the maximum is attained at a unique point and undefined when it is not. As a consequence, the usual $\sqrt{m}$–scaling does *not* yield a Gaussian limit for the centered statistic $\sqrt{m}\,\big\{T_{\max}(X_{1:m}) - T_{\max}(P)\big\}$; instead, after a different (typically linear-in-$m$) rescaling one obtains a non-Gaussian extreme-value limit law. Thus Assumption 2(iii) fails. Using max-pooling inside a deep encoder therefore breaks the loss-CLT of Theorem 2. (Softmax pooling with fixed temperature $\tau > 0$, by contrast, is smooth and can be made into a valid AL pooler.)

The table below summarises the status of common poolers.

| Pooler | AL / CLT? | Influence fcn. $\psi_P$ in $L^2(P)$? |
| --- | --- | --- |
| Sample mean | ✓ | ✓ |
| Huber $M$-estimator ($\delta$ fixed) | ✓ | ✓ |
| Sample median | ✓ | ✓ |
| Top-$k$ or max | ✗ | ✗ |
| Softmax ($\tau > 0$ fixed) | ✓ | ✓ |

**Smooth Approximation of Non-Regular Statistics**   The theory developed here establishes that Hadamard differentiability of the encoder ensures asymptotic normality and consistency, and in Appendix C.1 we develop the idea that our encoders learn sufficient statistics. But what if the sufficient statistic of interest is not Hadamard differentiable? The sample maximum is a classic example: it is the minimal sufficient statistic for the endpoint of a uniform distribution (see Example 2), yet it is not asymptotically normal.

Let $X_1, \ldots, X_m \sim \mathrm{Uniform}(0, \theta)$. The sample maximum

$$X_{(m)} := \max\{X_1, \ldots, X_m\}$$

satisfies

$$m(\theta - X_{(m)}) \xrightarrow{d} \mathrm{Exp}(1/\theta),$$

so it converges to $\theta$ but its asymptotic distribution is exponential, not Gaussian. This occurs because the maximum is not a smooth functional of the empirical distribution: it fails Hadamard differentiability, so the functional delta method does not apply.

A natural remedy is to approximate the max by a smooth, duplication-invariant functional. A standard choice is the *normalised log-sum-exp*:

$$\mathrm{LSE}_\lambda(X_1, \ldots, X_m) := \frac{1}{\lambda} \log\left(\frac{1}{m} \sum_{i=1}^m e^{\lambda X_i}\right).$$

For fixed $\lambda$, this is a smooth functional of the empirical measure (under mild moment conditions ensuring the log-moment is finite) and is therefore amenable to the delta-method theory above. As $\lambda \to \infty$, $\mathrm{LSE}_\lambda \to \max_i X_i$, so we recover the max in the limit.

**Corollary 3** (Smooth approximation suffices for asymptotic normality). *Let $T(P_m)$ be a non-smooth statistic (e.g., the maximum), and let $T^{(\lambda)}(P_m)$ be a family of smooth approximations (e.g., $\mathrm{LSE}_\lambda$) such that $T^{(\lambda)}(P_m) \to T(P_m)$ pointwise as $\lambda \to \infty$. Suppose that for each fixed $\lambda$ the map $P \mapsto T^{(\lambda)}(P)$ is Hadamard differentiable and satisfies Assumption 2. Then for any fixed $\lambda$, $T^{(\lambda)}(P_m)$ admits asymptotically normal plug-in estimators. Allowing $\lambda = \lambda_m \to \infty$ introduces a tradeoff between approximation error and $\sqrt{m}$-asymptotics.*

Thus, even when the true sufficient statistic is not regular, a Hadamard differentiable encoder can still be learned to approximate it. This ensures that the asymptotic guarantees from Theorem 2 continue to hold. This also highlights why we cannot use max-pooling in the encoder, since it breaks the $\sqrt{m}$ CLT.

**Generators**   All neural generators considered in the experiments—MLPs and Transformer decoders directly, and diffusion/score models when implemented with a fixed-step sampler—can be viewed as finite-dimensional compositions of smooth maps from latent codes to synthetic samples, inducing a (locally) smooth dependence of the resulting law on the embedding; this is the modelling assumption captured by Assumption 3.

### D.3   Embeddings and Predictive Sufficiency

**Setting.**   Let $\mathcal{M} \subset \mathcal{P}(\mathcal{X})$ be the statistical manifold introduced in Section 5.

Here we assume the statistical manifold $\mathcal{M}$ is $d$–dimensional (in the usual differential-geometric sense), so $\dim T_P \mathcal{M} = d$ for every $P \in \mathcal{M}$.

For $P \in \mathcal{M}$ observe $S_m = (X_1, \ldots, X_m) \overset{\text{i.i.d.}}{\sim} P$ and write the empirical measure $P_m = m^{-1} \sum_{i=1}^m \delta_{X_i}$.

Throughout we use the *plug-in predictor* $P_m$. Given a statistic $T_m = \phi(P_m)$, where $\phi : \mathcal{P}(\mathcal{X}) \to \mathbb{R}^d$ is defined and $C^1$ on a neighbourhood of $\mathcal{M}$, let $U \subset \mathbb{R}^d$ be an open set with $\phi(\mathcal{M}) \subset U$ such that $\mathbb{P}(\phi(P_m) \in U) \to 1$ for every $P \in \mathcal{M}$. Define a *reconstruction* map $R : U \to \mathcal{M}$ that is $C^1$ on $U$ (in the manifold sense) and set

$$P_m^\phi := R(T_m) = R(\phi(P_m)).$$

**Definition 7** (Predictive sufficiency). *The statistic $T_m = \phi(P_m)$ is asymptotically predictive sufficient if there exist an open set $U \subset \mathbb{R}^d$ with $\phi(\mathcal{M}) \subset U$, and a reconstruction map $R : U \to \mathcal{M}$ that is $C^1$ on $U$ (in the manifold sense), such that for every $P \in \mathcal{M}$,*

$$\mathbb{P}_{P^{\otimes m}}\big(\phi(P_m) \in U\big) \to 1 \quad \text{and} \quad \big\|P_m - R(\phi(P_m))\big\|_{\mathrm{BL}} \xrightarrow[m \to \infty]{P^{\otimes m}} 0.$$

We write $P_m^\phi := R(\phi(P_m))$.

This notion of sufficiency is a reconstruction-based asymptotic analogue of the "no order/size artifacts" principle in Section D.1: it asks that from the low-dimensional coordinate $\phi(P_m)$ one can reconstruct the plug-in predictor $P_m$ in the bounded–Lipschitz metric (and hence recover all weakly continuous predictive functionals).

**Theorem 3** (Embedding $\Longleftrightarrow$ Predictive sufficiency). *Assume that $\phi : \mathcal{P}(\mathcal{X}) \to \mathbb{R}^d$ is $C^1$ on a neighbourhood of $\mathcal{M}$ and satisfies the encoder regularity conditions of Assumption 2. Then the following are equivalent.*

    *(i)* Smooth embedding: *the restriction $\phi|_\mathcal{M} : \mathcal{M} \to \mathbb{R}^d$ is injective and its differential $d\phi_P : T_P\mathcal{M} \to \mathbb{R}^d$ is bijective for every $P \in \mathcal{M}$.*

    *(ii)* Predictive sufficiency: *$T_m = \phi(P_m)$ is asymptotically plug-in sufficient in the sense of Definition 7.*

*Proof (sketch).* Throughout, $\|\cdot\|_{\mathrm{BL}}$ denotes the bounded–Lipschitz norm on signed measures.

**(i)** $\implies$ **(ii).** If $\phi|_\mathcal{M}$ is a smooth embedding, its image $\phi(\mathcal{M}) \subset \mathbb{R}^d$ is an embedded submanifold. By the inverse–function theorem and standard tubular-neighbourhood constructions, for each $P \in \mathcal{M}$ there exists a neighbourhood $V_P$ of $\phi(P)$ in $\mathbb{R}^d$ and a continuous map $R_P : V_P \to \mathcal{M}$ such that $R_P(\phi(Q)) = Q$ for all $Q \in \mathcal{M}$ with $\phi(Q) \in V_P$. Using a partition of unity we may glue these local inverses into a single continuous retraction $R : V \to \mathcal{M}$ defined on an open neighbourhood $V$ of $\phi(\mathcal{M})$ and satisfying $R(\phi(Q)) = Q$ for all $Q \in \mathcal{M}$.

Encoder regularity (Assumption 2) gives

$$\sqrt{m}\left\{\phi(P_m) - \phi(P)\right\} = \frac{1}{\sqrt{m}} \sum_{i=1}^m \psi_P(X_i) + o_P(1) \quad \text{in } \mathbb{R}^d,$$

so $\phi(P_m) \to \phi(P)$ in probability. Since $P_m \to P$ in $d_{\mathrm{BL}}$ almost surely, we have $\phi(P_m) \in V$ with probability tending to one and

$$P_m^\phi := R(\phi(P_m)) \xrightarrow[m\to\infty]{P^{\otimes m}} R(\phi(P)) = P$$

in the bounded–Lipschitz topology, by continuity of $R$. Combining this with $P_m \to P$ in $\|\cdot\|_{\mathrm{BL}}$ and applying the triangle inequality yields

$$\|P_m - P_m^\phi\|_{\mathrm{BL}} \le \|P_m - P\|_{\mathrm{BL}} + \|P_m^\phi - P\|_{\mathrm{BL}} \xrightarrow[m\to\infty]{P^{\otimes m}} 0,$$

which is precisely predictive sufficiency in the sense of Definition 7.

**(ii)** $\implies$ **(i).** Conversely, assume predictive sufficiency with $R \in C^1(U, \mathcal{M})$. Fix $P \in \mathcal{M}$. Since $P_m \to P$ in $\|\cdot\|_{\mathrm{BL}}$ a.s. and $\|P_m - R(\phi(P_m))\|_{\mathrm{BL}} \to 0$ in probability, we have $R(\phi(P_m)) \to P$ in probability. Encoder regularity implies $\phi(P_m) \to \phi(P)$ in probability, hence by continuity of $R$, $R(\phi(P)) = P$. Therefore $R \circ \phi|_\mathcal{M} = \mathrm{id}_\mathcal{M}$.

Differentiating the identity map on $\mathcal{M}$ and using the chain rule yields

$$dR_{\phi(P)} \circ d\phi_P = \mathrm{id}_{T_P\mathcal{M}}.$$

Thus $d\phi_P$ is injective for every $P \in \mathcal{M}$, and since $\dim T_P\mathcal{M} = d = \dim \mathbb{R}^d$, it is bijective. Moreover, $R|_{\phi(\mathcal{M})}$ is a continuous inverse of $\phi|_\mathcal{M}$, so $\phi|_\mathcal{M}$ is a smooth embedding. $\qquad\square$

**Remark** (Identifiability is automatic). *Because each $P \in \mathcal{M}$ already defines a unique predictive distribution, any statistic that is plug-in sufficient must be injective; no separate identifiability condition is required.*

# E  Extensions

## E.1  Extension to Multiscale Settings

In many applications, data is naturally organized across multiple scales. For example, we may observe distributions of samples at a fine scale (e.g., single cells), grouped into entities at a coarser scale (e.g., patients), which themselves may belong to larger groups (e.g., hospitals). More generally, we may observe hierarchical data in which each level exhibits internal distributional structure.

Our framework naturally extends to such multiscale settings. At each scale $s$, we observe a set of units indexed by $i = 1, \ldots, n^{(s)}$. Each unit $i$ at scale $s$ is associated with: a set of samples $S_{i,m}^{(s)} = \{x_{ij}^{(s)}\}_{j=1}^{m}$, drawn i.i.d. from a distribution $P_i^{(s)}$ and a higher-scale sample $x_i^{(s+1)} \in \mathcal{X}^{(s+1)}$, representing the corresponding entity at scale $s + 1$.

The lower-scale distributions $P_i^{(s)}$ are drawn i.i.d. from a meta-distribution $Q^{(s)}$ over $\mathcal{P}(\mathcal{X}^{(s)})$, while the higher-scale samples $x_i^{(s+1)}$ are drawn from $P_i^{(s+1)}$, where $P_i^{(s+1)} \sim Q^{(s+1)}$.

Each lower-scale set $S_{i,m}^{(s)}$ defines an empirical measure

$$P_{i,m}^{(s)} = \frac{1}{m} \sum_{j=1}^{m} \delta_{x_{ij}^{(s)}} \in \mathcal{P}_m(\mathcal{X}^{(s)}).$$

At each scale we learn: an encoder $\mathcal{E}^{(s)} : \mathcal{P}_m(\mathcal{X}^{(s)}) \to \mathbb{R}^{d_s}$ mapping lower-scale empirical distributions into latent space, an encoder $\mathcal{E}^{(s+1)} : \mathcal{X}^{(s+1)} \to \mathbb{R}^{d_{s+1}}$ mapping higher-scale samples into the corresponding latent space, and generators $\mathcal{G}^{(s)} : \mathbb{R}^{d_s} \to \mathcal{P}(\mathcal{X}^{(s)})$ and $\mathcal{G}^{(s+1)} : \mathbb{R}^{d_{s+1}} \to \mathcal{P}(\mathcal{X}^{(s+1)})$ at each scale.

To link adjacent scales, we introduce deterministic maps

$$f^{(s)} : \mathbb{R}^{d_s} \to \mathbb{R}^{d_{s+1}} \quad \text{and} \quad g^{(s)} : \mathbb{R}^{d_{s+1}} \to \mathbb{R}^{d_s},$$

which project embeddings upward and downward between latent spaces.

We jointly train to enforce: *Approximate identity* at each scale:

$$\mathcal{G}^{(s)}(\mathcal{E}^{(s)}(S_{i,m}^{(s)})) \approx P_i^{(s)}, \quad \mathcal{G}^{(s+1)}(\mathcal{E}^{(s+1)}(x_i^{(s+1)})) \approx P_i^{(s+1)},$$

and *co-embedding consistency*: the mapped lower-scale embedding $f^{(s)}(\mathcal{E}^{(s)}(S_{i,m}^{(s)}))$ should align with the higher-scale embedding $\mathcal{E}^{(s+1)}(x_i^{(s+1)})$ and vice versa via $g^{(s)}$.

Formally, we optimize objectives of the form:

$$L = \mathfrak{d}\big(P_i^{(s)}, \mathcal{G}^{(s)}(\mathcal{E}^{(s)}(S_{i,m}^{(s)}))\big) \tag{2}$$

$$+ \quad \mathfrak{d}\big(P_i^{(s+1)}, \mathcal{G}^{(s+1)}(\mathcal{E}^{(s+1)}(x_i^{(s+1)}))\big) \tag{3}$$

$$+ \quad \|f^{(s)}(\mathcal{E}^{(s)}(S_{i,m}^{(s)})) - \mathcal{E}^{(s+1)}(x_i^{(s+1)})\|^2 \tag{4}$$

$$+ \quad \|g^{(s)}(\mathcal{E}^{(s+1)}(S_{i,m}^{(s+1)})) - \mathcal{E}^{(s)}(x_i^{(s)})\|^2 \tag{5}$$

where $\mathfrak{d}$ is a divergence or distance (e.g., KL divergence, Wasserstein distance) defined by the generative model. One natural approach would be to let $f^{(s)}, g^{(s)}$ both be the identity, forcing the model to learn a co-embedding across scales. But this may be too rigid and we might prefer more flexilbity in practice.

This bi-directional coupling ensures that embeddings at adjacent scales are mutually predictive and geometrically aligned, while each scale individually satisfies distributional invariance and approximate identity. The framework naturally generalizes to hierarchies involving more than two scales by recursively composing the maps $f^{(s)}$ and $g^{(s)}$ across levels.

# F   Broader impacts

Generative distribution embeddings provide a general framework for modeling data across scales. They are broadly applicable to a wide variety of problems, including those with direct societal consequences, for example in healthcare. In these settings, it will be critical to consider any potential inequities induced by GDEs, as is the case for any modelling approach. Lastly, we acknowledge the environmental impact of this paper, which used nontrivial amounts of computational resources, estimated to be about 54kg $CO_2$.

