# OpenReview forum: "Generative Distribution Embeddings: Lifting autoencoders to the space of distributions for multiscale representation learning"
_NeurIPS.cc/2025/Conference — NeurIPS 2025 poster_

### Official Review · Reviewer_N5oL · 2025-06-25

**Clarity:** 3
**Significance:** 3
**Originality:** 3
**Rating:** 4
**Confidence:** 4

**Summary:**

This paper studies a heirarchical perspective on data analysis and encoder-generator mechanisms in data science. The authors propose *generative distribution embeddings*, or GDEs, which appropriately lift the usual notion of encoder-generator onto the space of distributions. This is akin to many works that are pursuing a similar trend, which the authors discuss in the related work. The premise of this work is the following: the authors work with samples \{x_{ij}\}_{j=1}^m \sim P_i which is a high-dimensional distribution, and and P_i \sim Q for i = 1,\ldots, n, where $Q$ is a distribution over probability distributions. The goal is to learn an encoder-generator (E-G) from the samples, which they train using standard divergence-like frameworks, and suggest that essentially any existing scheme falls into their framework. They discuss how their approach is related to learning over statistical manifolds, as they relate the learning of encoders to sufficient statistics. They conclude with a suite of numerical experiments largely focused around computational biology, where their method appears to perform very well compared to existing approaches.

**Questions:**

- Lines195-199: Such a property (preservation of geodesics of distributions over distributions) is not guaranteed right?

- what is the significance of "2 bits" in section 6.2 (rather, maybe just talk about what this is measuring)

- section 6.3: why is the baseline a simple linear model?

- Section 6.4: I might be missing something but it seems like there are no other comparisons?

- at the start of section 2, maybe mention that $ i = 1,\ldots, n$ is the indexing over $P_i$

- caption of fig2: "PCs" is not defined --- I assume this means principal components, but this figure is just not clear in my opinion.

- I also insist on somewhat restructuring the main written portion of the article to increase clarity of the paper.

- line 101: extra ")" after $\epsilon_k$ I believe

**Ethical Concerns:**

["NO or VERY MINOR ethics concerns only"]

**Final Justification:**

The authors addressed all of my questions and re-wrote Section 2, which I had the most qualms with.

**Quality:**

3

**Strengths And Weaknesses:**

The main strength of this paper is in the novelty of performing encoding-decoding-type schemes for distributions over distributions. The connections to the Wasserstein space is also quite nice. The figures help guide the reader throughout the paper, and the experiments appear quite thorough, though I am far from an expert.

The main weakness of the paper is the writing---it is very verbose and seems rushed. In particular, there are many typos and a few places with repetitive sentence structure. For example, lines 189-194 are basically repeats of the same sentence twice. Proposition 3 is included between lines 211 and 212 with basically no context or follow-up; proposition 1 is similar (proposition 2 is motivated from the text).

---

> ### Author Rebuttal · Authors · 2025-07-31
>
> Thank you for this insightful review.
>
> > I also insist on somewhat restructuring the main written portion of the article to increase clarity
>
> Thank you for this suggestion. We propose the following changes:
>
> 1. Moving Sec. 6.4 to the Appendix to allow more room for exposition.
> 2. Replacing Sec. 2 with the revised version below. The content remains the same, but with clearer exposition.
> 3. Adding a new section "Implementation recipe for GDEs" after Sec. 3., or in the Appendix depending on space (see reviewer eYba response for proposed text).
>
> > section 6.3: why is the baseline a simple linear model?
>
> For the perturbation response prediction task and dataset considered in Sec. 6.3, simple linear models are state-of-the-art and outperform deep learning-based alternatives such as scGPT and GEARS [AHA24, WWS+25].
>
> We have now added another baseline method, the VAE-based scVI [LRC+18]. To adapt scVI to distributional tasks, we use the mean of sample-wise scVI embeddings as a distribution embedding. Again, we find that GDEs outperform scVI, which achieves $R^2 = 0.42$ and $\text{MSE}=1.55$ (compare with 0.46 and 1.50 respectively for GDEs).
>
> [AHA24] Ahlmann-Eltze, C., Huber, W., & Anders, S. (2024). Deep learning-based predictions of gene perturbation effects do not yet outperform simple linear baselines. BioRxiv.
>
> [WWS+25] Wu, Y., et al. (2025). PerturBench: Benchmarking machine learning models for cellular perturbation analysis. arXiv.
>
> [LRC+18] Lopez, R., et al. (2018). Deep generative modeling for single-cell transcriptomics. Nature methods.
>
> > Section 6.4: I might be missing something but it seems like there are no other comparisons?
>
> You are correct -- in this application we do not compare against any baseline. There are two practical reasons for this: (1) there are, to our knowledge, no existing generative models for predicting the morphological response of novel perturbations in optical pooled screen data (2) the scale of the dataset (30M images) precludes easy training of baseline methods.
>
> Our main goal here is to show that GDEs are compatible with image data and can generate realistic images (see Fig. 7) with relevance to an open problem in biology. We do not claim state-of-the-art performance for this particular task. As such, we will move this section (6.4) to the appendix, and use the space for clearer exposition.
>
> At the reviewer's suggestion we are training a variant of our GDE implementation using a VAE as the generator for the revised manuscript, and we are happy to consider other baselines.
>
> > what is the significance of "2 bits" in section 6.2 (rather, maybe just talk about what this is measuring)
>
> Here, "bits" refers to the amount of predictive information ($t_1 \to t_2$) contained in the representations. We will clarify this and provide intuition. At line 283, we will add:
>
> "This suggests that GDE representations enable more granular prediction of cell fates. If cell fates were discrete and uniformly likely, 1 bit of mutual information would correspond to resolving 2 fates while 3 bits would correspond to resolving 8 fates."
>
> > Lines195-199: Such a property (preservation of geodesics of distributions over distributions) is not guaranteed right?
>
> Correct, this is an empirical phenomenon that we observe in this particular setting. In fact the geodesics will "adapt" to the data and the prior over $\mathcal M$ as we show in Section 5.3. We will emphasize this in the revised manuscript.
>
> > caption of fig2: "PCs" is not defined [...] this figure is just not clear in my opinion.
>
> Yes, PCs refers to principal components. We will rewrite the caption:
>
> "**Figure 2: Concentration of distribution embeddings and plug-in loss.** (Left) Distribution of plug-in GDE loss (with diffusion generator) for MNIST image sets of different sizes sampled from the same distribution $P_i$ over MNIST digit identities. (Right) First two principal components of 64D embeddings of sample sets of different size generated by the same $P_i$".
>
> > Proposition 3 is included between lines 211 and 212 with basically no context or follow-up; proposition 1 is similar (proposition 2 is motivated from the text).
>
> Thank you for highlighting this. We have now entirely rewritten Sec. 2 (see below), addressing this comment.
>
> > There are many typos and a few places with repetitive sentence structure. For example, lines 189-194 are basically repeats of the same sentence twice.
>
> We apologize. We will remove lines 192-194 and carefully correct typos in the revised manuscript.
>
> > at the start of section 2, maybe mention that $i=1,\dots, n$ is the indexing over $P_i$
>
> Thank you -- we have addressed this in the revised Section 2.
>
> > line 101: extra ")" after $\epsilon_k$
>
> Thank you for catching this. It will be removed.
>
> ## Revised Section 2
> ## 2 Setting and methods
> ### 2.1 Motivating example
> Many modern datasets consist of several large groups of exchangeable samples: for example, single cells or DNA sequences collected from individual patients. Suppose we observe $n$ such groups,
>
> $$
>   \mathcal D = \bigl\\{S_{i, m} = \\{x_{ij}\\}^m_{j=1} \bigr\\}^n_{i=1}, \qquad x_{ij} \in \mathcal X
> $$
>
>
> where each group $S_{i,m}$ is drawn i.i.d. from an unknown distribution $P_i \in \mathcal{P}(\mathcal{X})$, and the distributions themselves are drawn from a meta-distribution $Q$:
>
> $$P_i \stackrel{iid}{\sim} Q,\qquad x_{ij} \stackrel{iid}{\sim} P_i.$$
>
> This is a classical hierarchical data generating process, which gives rise to a multiscale problem. The subject of interest here is not only individual cells or DNA sequences $x_{ij}$, but the patients $P_i$. As we explore in Sec. 3 and concretely demonstrate in Sec. 6, this setting is broadly applicable beyond these particular examples.
>
> In practice, there are often two major challenges in modeling this kind of data. First, unit-level data is often inherently noisy. For example, single-cell data suffers from noise due to molecular undersampling [39]. Our goal is to learn patient embeddings which capture distribution-level signal, rather than the sample-level noise. The second challenge is that groups can contain millions of samples (in the case of DNA sequencing reads per patient, $m$ can be $\sim10^8$). It is computationally infeasible to train a model on all samples simultaneously, but given the inherent noise at the unit level we would benefit from embedding all available samples at inference time.
>
> In the remainder of Sec. 2, we will show that both of these practical challenges can be overcome by learning distribution embeddings rather than simply encoding sets of samples. The distributional perspective enables models which distill distribution-level signal, and are able to massively scale at inference time to make use of all available data for precise embeddings.
>
> ### 2.2 Learning generative distribution embeddings (GDEs)
> We address this problem with GDEs, which consist of an encoder $\mathcal E$ that maps a finite set of samples $S_{i,m}$ to a latent representation, and a conditional generator $\mathcal G$ that maps the latent representation back to the sample space. Formally, we aim to learn $\mathcal{E}, \mathcal{G}$ such that
>
> $$\mathcal G\bigl(\mathcal E(S_{i,m})\bigr)\ \stackrel{m \to \infty}{\longrightarrow}\ P_i\tag{1}$$
>
> The training procedure is shown in Algorithm 1:
> ```
> For each minibatch of sets $S_{i,m}$:
>     Compute latent representation $z_i = \mathcal E(S_{i,m})$.
>     Compute generator loss $\ell(P_i, \mathcal G(z_i))$.
>     Backpropagate gradients through $\mathcal E$ and $\mathcal G$.
> End for.
> ```
>
> The loss $\ell$ is the standard training objective for the conditional generator (for example, an evidence lower bound for a VAE or a denoising score-matching objective for a diffusion model); we do not need to backpropagate through the sampling process of $\mathcal G$.
>
> ### 2.3 Distributional invariance uniquely enables achieving Eq. (1)
>
> We show that achieving Eq. (1) is only possible if the encoder satisfies the following two constraints (formal proofs in Appendix D):
> 1. Permutation invariance: reordering the samples in $S_{i,m}$ does not change the embedding.
> 2. Proportional invariance: duplicating every sample $K$ times ($\bigcup_{k=1}^K S_{i, m}$) does not change the embedding.
>
> We refer to an encoder with these properties as *distributionally invariant*: the encoder must depend only on the empirical distribution. So for some function $\phi$ we can write:
> $$P_{i, m} = \frac{1}{m} \sum_{j=1}^m \delta_{x_{ij}},
> \qquad
> \mathcal E(S_{i,m}) = \phi(P_{i,m}).$$
>
> We show formally that distributionally invariant encoders can capture any distributional property and furthermore that any non-distributionally invariant architecture can spuriously encode noise features irrelevant to the distribution:
>
> [Proposition 1 box]
>
> Beyond separating signal and noise, distributional invariance has a second crucial consequence: it enables a *central limit theorem for the embeddings*. As the set size grows, $\mathcal E(S_{i,m})$ concentrates around its population value with Gaussian fluctuations:
> $$\sqrt{m}\ \bigl(\mathcal E(S_{i,m})-\phi(P_{i,m})\bigr) \ \xrightarrow{d}\  \mathcal N(0,\Sigma)$$
>
> This result, illustrated empirically in Fig. 2, is what makes encoding massive sets possible.
>
> Because the embeddings converge, the plug-in loss
> $$\widehat\ell_m = \ell\bigl(P,\mathcal G(\mathcal E(S_{i,m}))\bigr)$$
>
> is an unbiased, asymptotically normal estimator of the population loss. This provides theoretical justification for training GDEs on subsets of larger sample sets: the gradient of $\widehat\ell_m$ computed on small sets matches (in expectation) the gradient computed using all samples per set. We prove this in Appendix D.2:
>
> [Proposition 2 box]
>
> Violating distributional invariance (for example, by using sum pooling) causes the embedding to depend on set size and breaks this limit theory causing Eq. (1) to fail.  In contrast, **mean pooling and differentiable M/Z-estimators** satisfy these properties.

---

> > ### Comment · Reviewer_N5oL · 2025-08-04
> >
> > Thank you for the detailed rebuttal and re-write of Section 2. I have decided to raise my score as a result.

---

### Official Review · Reviewer_eYba · 2025-07-03

**Clarity:** 2
**Significance:** 3
**Originality:** 3
**Rating:** 5
**Confidence:** 2

**Summary:**

This paper introduces Generative Distributional Embeddings framework for learning representations for distributions based on an _distributionally invariant_ encoder and conditional generative decoder.

**Questions:**

- A suggestion: a "recipe" for building a distributionally invariant encoder and prefix-based conditional generation would be very useful. I see the details present in the supplementary material but bringing it as its own section (separate from the theoretical results) and a figure/table summarizing encoder/decoder architectures for different experiments would make it more accessible.

**Ethical Concerns:**

["NO or VERY MINOR ethics concerns only"]

**Limitations:**

yes

**Quality:**

4

**Strengths And Weaknesses:**

## Strengths

1. Clear theoretical contribution in terms of "distributionally invariant" encoder and interpretation as a asymptotic predictive sufficient statistics.
2. Positions prior work clearly in context of this framework.
3. Clear demonstration in several computational biology problems.

## Weaknesses

- I did not check the theoretical results in detail
- Presentation is dense - I understand showing all the six applications in main text is good but given that this is a strong submission, I would remove some of the experimental section to explain the theoretical contribution more clearly. I read a Wasserstein Wormhole and a couple of other references to understand better.

---

> ### Author Rebuttal · Authors · 2025-07-31
>
> Thank you for this thoughtful review.
>
> > Presentation is dense - I understand showing all the six applications in main text is good but given that this is a strong submission, I would remove some of the experimental section to explain the theoretical contribution more clearly.
>
> Thank you for this suggestion. We are now moving Sec. 6.4. to the Appendix. We will use the extra space to include a revised Sec. 2 with clearer exposition (with the same content), see response to reviewer N5oL for text.
>
> > A suggestion: a "recipe" for building a distributionally invariant encoder and prefix-based conditional generation would be very useful. I see the details present in the supplementary material but bringing it as its own section (separate from the theoretical results) and a figure/table summarizing encoder/decoder architectures for different experiments would make it more accessible.
>
> Thank you for this suggestion. In our revision, we will add a dedicated section summarizing "recipes" for constructing GDEs, including a table that enumerates the encoder and generator pairings used in each of our experiments, to enable easy implementation of GDEs. This addition will be included either in the Appendix or after Section 3, space permitting. See our proposed addition below:
>
> ---
>
> ## Implementation recipe for GDEs
>
> The GDE framework is instantiated by pairing a distributionally invariant encoder with a conditional generative model. The following steps outline a general recipe for building GDEs across diverse data domains:
>
> 1. **Sample from the metadistribution (construct sets)**
>    Group raw data into sets $S_i = \\{x_{ij}\\}_{j=1}^m$, where each set reflects a draw from an unknown latent distribution $P_i$. Groupings can be based on discrete metadata (e.g., text by author, reviews by rating, images by label, cell clones, gene perturbations) or continuous metadata (e.g., time, location, expression quantiles). Sets need not be mutually exclusive, meaning a single data point can belong to multiple sets.
>
> 2. **Choose a distributionally invariant encoder**
>    Select or construct a distributionally invariant encoder $\mathcal{E}$. This selection generally involves (1) using an architecture for element-wise embeddings and (2) pooling across element-wise embeddings with a sample mean (or other M-estimate). We found that architectures with multiple pooling layers, where each layer's pooled output is concatenated with the element-wise embeddings, were particularly effective. This contrasts with pure DeepSets-style architectures that only pool once at the final layer. For deeper architectures, we have found that including skip-connections improves performance, especially if the generator is also a relatively deep network.
>
> 3. **Build a conditional generator**
>    The generator $\mathcal{G}$ "decodes" from latent space back to the sample space. It should be conditionable on $z = \mathcal{E}(S)$.
>
> 4. **Train via plug-in loss**
>    Optimize the generator to minimize the generator loss function $\ell(P_m, \mathcal G(\mathcal E(S_m)))$. This loss should be the standard training objective for the conditional generator. This plug-in loss encourages reconstruction of the true distribution.
>
> The encoder-generator pairs used for each application in the paper are shown in Table 3.
>
> ### Table 3: Encoder and Generator Architectures by Experiment
>
> | Section | Task | Set Construction | Encoder Architecture | Generator Architecture | Notes |
> |--------|------|--------|-----------------------|--------------------------|-------|
> | 6.1 | MNIST, FMNIST | Same image class | see Table 1 | see Table 1 | Synthetic data benchmark |
> | 6.2 | Lineage-traced scRNA-seq |  Same cell clones |  ResNet-GNN | CVAE |  |
> | 6.3 | Genetic perturbation (scRNA-seq) |  Same perturbation |  ResNet-GNN | CVAE |  |
> | 6.4 | Morphological responses (cell images) |  Same perturbation |  2D Conv-GNN | DDPM (U-Net) |  |
> | 6.5 | Tissue-specific methylation |  Same patient; Same tissue type |  1D Conv-GNN | HyenaDNA |Uses prefix conditioning |
> | 6.6 | Yeast promoter quantile decoding |  Expression quantile (continuous) |  1D Conv-GNN | HyenaDNA | Uses prefix conditioning |
> | 6.7 | Viral protein spatiotemporal modeling |  Same sampling month and location |  ESM + mean pooling | ProGen2 | Uses prefix conditioning |

---

### Official Review · Reviewer_45Ka · 2025-07-06

**Clarity:** 4
**Significance:** 4
**Originality:** 4
**Rating:** 5
**Confidence:** 2

**Summary:**

The paper presents Generative Distribution Embeddings (GDE), a framework for learning representations of probability distributions by combining distribution-invariant encoders with conditional generative models. The authors formalize the concept of distributional invariance, building on empirical process theory and information geometry, and connect GDEs to predictive sufficiency and statistical manifold geometry. The proposed method is benchmarked in low dimensional synthetic data and high dimensional structured domains including single cell omics, DNA methylation, protein sequence modelling and image based phenotyping, each presenting unique challenges for distribution level modelling. The framework leverages modern generative models and suports flexible grouping strategies to enable application to several scientific questions.
Empirically, GDEs outperform kernel mean embeddings and Wasserstein Wormholes, and can recover Wasserstein geometry in latent space even in for high dimensional data. The main contributions are the unification of prior approaches, empirical demonstration of geometric properties, and broad applicability.

**Questions:**

Questions and comments provided in addition to what was mentioned in the strengths and weaknesses.
- The formalization of "distributional invariance" is necessary for the framework, but it is not clear that this property is sufficient for learning useful or interpretable representations in practice, especially when the meta-distribution Q is misspecified or when data are not IID. Can the authors provide theoretical or empirical evidence for the robustness of GDEs to such misspecification, and clarify whether distributional invariance alone is enough for practical utility?
- The paper positions GDEs as a unifying framework and demonstrates empriical advantages over baselines. Part of the novelty is, indeed, in the synthesis, as well as the breadth of application. I think the authors should do a bit more to clarify the necessity and sufficiency of the formalization: what are the concrete advantages of the GDE abstraction over existing methods (e.g., kernel mean embeddings, Neural Statistician, flow matching) in practical settings?

**Ethical Concerns:**

["NO or VERY MINOR ethics concerns only"]

**Final Justification:**

i have read the rebuttals and the other reviews of this paper. I am happy to maintain my current score.

**Limitations:**

yes

**Quality:**

4

**Strengths And Weaknesses:**

**Strengths:**
- Rigorous formalization of distributional invariance, with clear connections to foundational work in empirical process theory and information geometry, and grounding the framework in predictive sufficiency, a property supporting that the learned representations capture the essential structure of distributions for downstream inference.
- Empirical demonstration that GDEs can recover Wasserstein geometry in latent space for simple distributions, which is concrete supporting evidence for the presented theoretical claims.
- General framework, allowing any distributionally invariant encoder and conditional generator and grouping strategies, and leveraging advantages in modern generative modelling to address the distribution level inference across scientific domains. The success of this framework depends on making sensible, domain informed choices for encoder, generator, and grouping strategies. The paper demonstrates this by benchmarking on several downstream tasks involving high dimensional biological data, adapting architectures and grouping to each domain, and showing strong results in each case.
- Limitations are explicitly acknowledged and discussed (in particular with what concerns the curse of dimensionalty of Wasserstein metrics that are central to this work), and the manuscript is generally clear.

**Weaknesses:**
- The evaluation metrics (Wasserstein, Sinkhorn) are less informative in high-dimensional spaces. The paper does mention this, but in the evaluation does not include alternative metrics or qualitative analyses.
- On interpretability: the method is benchmarked on several computational biology problems. This is not the main focus of the paper, but it would have been valuable to see more analysis of interpretability/biological relevance to close the loop.
- Furthermore, the impact of grouping strategies and prior selection (meta-distribution Q) is not systematically analyzed. The choices for this component can lead to uninterpretable or misleading representations, so I would have liked to see some discussion on this. Indeed, this could provide further principled guidance on reproducibility to practitioners, which I think could be better addressed by the paper.

---

> ### Author Rebuttal · Authors · 2025-07-31
>
> Thank you for this thoughtful review.
>
> > The evaluation metrics (Wasserstein, Sinkhorn) are less informative in high-dimensional spaces. The paper does mention this, but in the evaluation does not include alternative metrics or qualitative analyses.
>
> Thanks for this observation. We now address this issue for our high-dimensional benchmarks (MNIST, FMNIST) by (1) including Frechet distance in pretrained Resnet18 embeddings (akin to the widely-used FID) as an evaluation metric and (2) including visualizations of generated images in our benchmarks. Both additional results align with the results provided in our existing benchmark: GDEs outperform existing methods (Wasserstein Wormhole and KME). We provide the FID results below. While we are not able to provide the generated images (due to new response format), we will include these in the appendix of our revised manuscript. The GDE generated images are clearly visually closer to the ground truth than Wasserstein Wormhole or coupled KME+DDPM.
>
> Frechet distance in Resnet18 representation space for image experiments from Table 2:
> | Method        | MNIST  | FMNIST  |
> |---------------|------|------|
> | GDE           | 103.5 | 132.4 |
> | KME+DDPM      | 110.2 | 146.9 |
> | W2 Wormhole   | 524.05 | 656.1 |
>
> > On interpretability: the method is benchmarked on several computational biology problems. This is not the main focus of the paper, but it would have been valuable to see more analysis of interpretability/biological relevance to close the loop.
>
> We appreciate this feedback. We have now more carefully interpreted the results of our clonal representation learning model (Sec. 6.2). In particular, we examine the clones for which GDE fate predictions are high quality, while Wasserstein Wormhole predictions are poor quality (as quantified by the difference in point-wise  mutual information, $\Delta \text{pMI}$). Of the 10 clones with largest $\Delta \text{pMI}$, 8 clones have multiple differentiation fates (e.g., both Neutrophil and Monocyte fates). This observation suggests that performance improvements are due to GDEs more effectively capturing complex fates (combination of two cell types). We will include these results in the appendix of our revised manuscript.
>
> Below, we also make some informal observations about biological relevance which we will include in Appendix B:
>
> - **Perturbation prediction in optical pooled screening data.** The genes for which GDEs most accurately predict perturbation responses are typically related to cell cycle (e.g., DONSON). This suggests that GDEs have learned representations which capture nuclear features related to abnormal cell cycle progression.
>
> - **Spatiotemporal modeling of viral lineage distributions.** In this application we find that  GDE representations are organized primarily by time and contain a much weaker spatial signal. This is in line with the intuition that there is significantly lower spatial variation in viral protein sequences than variation in time, due to fast transmission.
>
> > Furthermore, the impact of grouping strategies and prior selection (meta-distribution Q) is not systematically analyzed. The choices for this component can lead to uninterpretable or misleading representations, so I would have liked to see some discussion on this. Indeed, this could provide further principled guidance on reproducibility to practitioners, which I think could be better addressed by the paper. (From below) ... [How does the encoder work] when the meta-distribution Q is misspecified or when data are not IID
>
> We agree that the choice of metadistribution has a significant impact on the learned representations. We demonstrate this quantitatively in Sec. 5.3, Fig. 5, where  we show that representations can be stretched or concentrated in certain regions depending on choice of metadistribution $Q$.
>
> As you correctly observe, this introduces a critical implementation choice for practitioners. However, we view this consideration as a key feature of the method -- GDEs can be adapted to learn task-specific geometries by adjusting the metadistribution $Q$.
>
> We will emphasize this point and provide suggestions for practical choices of metadistribution in the "recipe" we are adding to the revised manuscript (text included in response to reviewer eYba).
>
> > The formalization of "distributional invariance" is necessary for the framework, but it is not clear that this property is sufficient for learning useful or interpretable representations in practice... Can the authors provide theoretical or empirical evidence for the robustness of GDEs to such misspecification, and clarify whether distributional invariance alone is enough for practical utility?
>
> We agree that this is an important point to clarify.
>
> There are two key insights in Thm 1: first as the reviewer notes, we show that the encoder cannot include more information than the empirical distribution; the flip side of the theorem proves that for IID data a distributionally invariant encoder can, in principle, express any property of the underlying distribution. In practice we find that this expressivity is very effective, and across all of our applications (Secs. 6.1–6.7) an empirical-measure based encoder learns representations that are useful and interpretable.
>
> When data within a set are not IID (for example due to temporal, spatial, or network dependence) the situation changes. In these cases the empirical distribution is no longer the correct invariant object, so an encoder that simply pools IID samples is not sufficient. Instead, one must adapt the notion of invariance to respect the structure of dependence. For example:
> - for time series or sequential data, one could pool over blockwise empirical distributions or use causal attention mechanisms;
> - for spatial or network data, one could pool over neighborhood empirical distributions or graph orbits.
>
> Once the correct invariant is used, the same arguments underlying Thm. 1 and Thm. 2 can be extended using tools such as Stein’s method for dependency graphs and martingale CLTs, which allow analogous guarantees under suitable mixing or dependence conditions.
>
> In short, for IID data Thm. 1 shows that distributional invariance is both necessary and sufficient for universal expressivity, and in practice we see that this simple approach already works well. For more structured dependence, the same principle applies after modifying the invariant, and we view this as an important and exciting avenue for future work. **To clarify these points we will include an additional section in Appendix D that outlines these issues and sketches a general approach to extend the core GDE results to dependent data.**
>
> All this being said, in our viral lineage application (Sec. 6.2) the data is clearly temporally and spatially dependent, yet performance remains reasonably strong, suggesting that modest violations of IID do not totally degrade results.
>
> > The paper positions GDEs as a unifying framework and demonstrates empriical advantages over baselines. Part of the novelty is, indeed, in the synthesis, as well as the breadth of application. I think the authors should do a bit more to clarify the necessity and sufficiency of the formalization: what are the concrete advantages of the GDE abstraction over existing methods (e.g., kernel mean embeddings, Neural Statistician, flow matching) in practical settings?
>
> GDEs strictly generalize existing approaches like KMEs, Neural Statistician, and the Wasserstein Wormhole, providing a systematic framework for combining different encoders and generators while previous works have focused on particular architectures (like the kernels, VAEs, or transformers). For example, GDEs show how we can use diffusion/flow matching as a distribution generator. This flexibility pays clear dividened in our synthetic benchmark (Sec. 6.1), where we show that GDEs more closely recover data distributions than existing alternatives.
>
> Beyond this generic flexibility and the performance it comes with, a core practical contribution of GDEs is that they enable leveraging existing domain-specific architectures for sample-level modelling into distribution-level representation learning models. This insight is particularly important because while we have many domain-specific modelling approaches at the level individual datapoints (for sequences, images, proteins, etc.) we have relatively few domain-specific approaches for learning representations of distributions.
>
> GDEs allow one to immediately build on domain-specific architectural innovations by connecting conditional generative models to distributionally-invariant encoders (e.g., a sample level encoder with mean-pooling).
>
> A clear illustration of this utility is the protein modeling task (Sec. 6.7): here, we repurpose the ESM and Progen2 architectures, which were not designed for distribution-level tasks, to learn representations of spike protein distributions. This repurposing would not be possible with existing approaches which lack the flexibility to integrate domain specific architectures.

---

> > ### Author Response · Authors · 2025-08-07
> >
> > Thank you again for your thoughtful review. In line with the updated guidance for the discussion period, we wanted to follow up to see if you have any remaining questions. As a reminder, the discussion period will end 11:59pm AoE on August 8th.

---

> ### Comment · Area_Chair_HyMA · 2025-08-07
>
> Dear Reviewer 45Ka,
>
> Just a kind reminder to please take a moment to review the authors’ rebuttal and see if it addresses your concerns, as the reviewer-author discussion deadline is approaching.
>
> Thank you again for your thoughtful and valuable input throughout the review process.
>
> Warm regards,
>
> The AC

---

### Official Review · Reviewer_aHtV · 2025-07-12

**Clarity:** 2
**Significance:** 3
**Originality:** 2
**Rating:** 5
**Confidence:** 4

**Summary:**

Generative Distribution Embeddings (GDE) is a framework that extends autoencoders to learn representations of entire probability distributions rather than individual datapoints. An encoder ingests a set of samples (an empirical distribution) and produces a latent code, while a conditional generator (in place of a traditional decoder) uses that code to generate new samples, aiming to match the original distribution. A key requirement is distributional invariance of the encoder – the representation should depend only on the underlying distribution and not on particular sample realizations. The authors provide a theoretical grounding, showing that GDE encodings behave as predictive sufficient statistics that capture the underlying distribution’s structure while abstracting away sampling noise. Notably, distances in the learned latent space correlate with 2-Wasserstein (W2) distances between distributions, and linear interpolations in latent space correspond to approximate optimal transport paths for simple distribution families. Empirically, GDE is benchmarked on synthetic distribution datasets, outperforming prior methods in reconstructing distributions. The paper further demonstrates GDE’s versatility on six diverse tasks in computational biology – from modeling cell populations and predicting single-cell perturbation responses to designing DNA sequences – showcasing strong performance and meaningful distribution-level embeddings across these domains.

**Questions:**

**Questions:**

- On line 76, shouldn't the notation be $S_{\cdot,k}$ to be consistent with the notation introduced earlier?
- When using a DDPM as a decoder, could you clarify how the training is performed? Specifically, do you backpropagate through the entire diffusion inference procedure?
- Could you provide a more detailed explanation of how the chosen architectures are modified or constrained to achieve distributional invariance?

**Some suggestions:**

- L70, it would be helpful to clearly define the input and output domains of the encoder and decoder, explicitly stating that: (i) The encoder takes as input a finite set of points (an empirical distribution) and outputs a latent embedding, (ii) The decoder takes this latent embedding as input and outputs samples forming a set of points, effectively reconstructing an empirical distribution. This clarification would facilitate understanding and immediately highlight the architecture’s core functionality.

- Section 3 is somewhat difficult to follow. It appears to describe how to construct the meta-distributions from labeled datasets, covering both discrete and continuous labels. However, the explanation—particularly for continuous labels—should be clarified to make the procedure more transparent and easier to understand.

- Line 78 is somewhat misleading and interrupts the reading flow. It doesn't seem particularly helpful since there is obviously a one-to-one correspondence between the sample set \(S_m\) and the empirical distribution \(P_m\). It might be clearer and more concise to omit or rephrase this detail.

-For the evaluation on the synthetic data benchmark, I'd recommend also computing the Sinkhorn divergence as a metric for the low-dimensional (i) Gaussian and (ii) Gaussian mixture settings. Given the low dimensionality ($d=5$), directly computing the Sinkhorn divergence in the ambient space is computationally feasible and would provide a richer comparison. Currently, the authors only compute the $W_2$ distance between Gaussian approximations of the empirical distributions (similar to the FID metric), which assesses only reconstruction accuracy for the first two moments.

**Ethical Concerns:**

["NO or VERY MINOR ethics concerns only"]

**Final Justification:**

Thank you for your thoughtful responses to my comments. I'm satisfied with the clarifications provided and have decided to increase my score to 5 accordingly.

**Limitations:**

Yes, the authors have included a dedicated section discussing the limitations of their approach.

**Paper Formatting Concerns:**

NA.

**Quality:**

3

**Strengths And Weaknesses:**

**Strengths.**
The proposed GDE framework is conceptually strong and general, unifying conditional generative modeling with representation learning for distributions. It subsumes or generalizes prior approaches like kernel mean embeddings and a recent “Wasserstein Wormhole” method under one umbrella. The theoretical contributions are substantial: the authors prove that under mild conditions GDE encoders learn asymptotically sufficient summaries of distributions, and they empirically confirm that the latent space geometry mirrors true distributional geometry (with latent Euclidean distances correlating highly with W2 distances and latent interpolation recovering transport geodesics). The experiments are extensive and convincing. On synthetic benchmarks (Gaussians, Gaussian mixtures, and image-defined distributions), GDE achieves lower reconstruction divergences than baselines (e.g. halving the Wasserstein error of a kernel mean embedding + DDPM and vastly outperforming the Wormhole approach). Furthermore, GDE’s practical impact is evidenced by strong results on real-world tasks – for instance, it improves the mutual information in predicting future cell states by over 2 bits compared to a baseline embedding, and it boosts the $R^2$ in zero-shot gene perturbation prediction (versus a mean-only predictor) from ~0.38 to ~0.46 while reducing error. These gains illustrate that GDE’s distribution-level representations are meaningfully more informative than simpler approaches.

**Weaknesses.**
The ambition of covering many scenarios comes with trade-offs. Some experimental evaluations feel a bit shallow or limited in baseline comparison; each biology application uses GDE in a novel context, but often the only point of comparison is either a naive approach (e.g. predicting only distribution means) or one specialized baseline (e.g. Wasserstein Wormhole), rather than a broader suite of competing methods. This makes it harder to pinpoint how universally superior GDE is. Additionally, the assumption of distributional invariance in the encoder design, while well-motivated, imposes a non-trivial constraint – not all standard set encoders qualify (e.g. simple DeepSets can be sensitive to sample size), so careful architectural choices (graph neural networks with pooling, attention mechanisms, etc.) are needed. The paper introduces new terminology and a combination of ideas from optimal transport, information geometry, and generative modeling, which means the exposition is quite dense; readers not already familiar with these backgrounds might struggle with clarity in places. Lastly, the method’s scope of novelty could be perceived as somewhat incremental (combining known building blocks) – the true innovation lies in integration and scale, rather than a fundamentally new algorithmic component, and this is something to keep in mind when weighing its contributions. Additionally, the core engineering and methodological components are not always clearly described. For instance, when using a DDPM as the decoder, it appears that the model is trained by backpropagating through the diffusion model's inference process. This effectively makes the training simulation-based and computationally demanding—an important detail that the authors do not explicitly address.

---

> ### Author Rebuttal · Authors · 2025-07-31
>
> Thank you for this insightful review.
>
> > Some experimental evaluations feel a bit shallow or limited in baseline comparison; each biology application uses GDE in a novel context, but often the only point of comparison is either a naive approach (e.g. predicting only distribution means) or one specialized baseline (e.g. Wasserstein Wormhole), rather than a broader suite of competing methods.
>
> We appreciate this feedback. While our synthetic benchmarks evaluate a broad range of methods, the reviewer is correct that we have not compared against large panels of methods for each of the applications in Sec. 6. We fully agree that it would be more satisfying to benchmark against several methods in each case. However, due to the scale of the datasets, comparisons to a large suite of competing methods may become impractical.
>
> In several cases we have compared against a state-of-the-art method. In Sec. 6.2, we use the Wasserstein Wormhole method which is state-of-the-art for cell set representations. In Sec. 6.3, a linear model for predicting perturbation response may appear naive, but in fact achieves state-of-the-art performance for the task [AHA24, WWS+25]. In Sec. 6.7, we include the widely-used ESM model as a baseline.
>
> In addition to these existing comparisons, we have now included the additional baseline of scVI [LRC+18], a VAE-based method, for our two single-cell transriptomic applications (Sec. 6.2, 6.3). In order to adapt scVI to distributional tasks, we use the mean of sample-wise scVI embeddings.
>
> We find in both cases that GDEs outperform scVI. For lineage tracing (6.2), scVI achieves 0.86 bits of mutual information -- lower than both Wasserstein Wormhole and GDEs. For perturbation prediction (6.3), scVI achieves $R^2 = 0.42$ and $\text{MSE}=1.55$, worse than GDEs but better than a pure linear model. We will add these results to the revised manuscript.
>
> [AHA24] Ahlmann-Eltze, C., Huber, W., & Anders, S. (2024). Deep learning-based predictions of gene perturbation effects do not yet outperform simple linear baselines. BioRxiv, 2024-09.
>
> [WWS+25] Wu, Y., Wershof, E., Schmon, S. M., Nassar, M., Osiński, B., Eksi, R., Yan, Z., Stark, R., Zhang, K., & Graepel, T. (2025). PerturBench: Benchmarking machine learning models for cellular perturbation analysis. arXiv.
>
> [LRC+18] Lopez, R., Regier, J., Cole, M. B., Jordan, M. I., & Yosef, N. (2018). Deep generative modeling for single-cell transcriptomics. Nature methods, 15(12), 1053-1058.
>
> > Additionally, the assumption of distributional invariance in the encoder design, while well-motivated, imposes a non-trivial constraint – not all standard set encoders qualify (e.g. simple DeepSets can be sensitive to sample size), so careful architectural choices (graph neural networks with pooling, attention mechanisms, etc.) are needed.
>
> The reviewer is correct that GDEs require careful choices for encoder architectures. In order to make the constraints clear for practitioners, we will include a "recipe" with clear guidelines constructing GDE architectures (see response to reviewer eYba), which shows which pooling functions are permissible under the constraint of distributional invariance.
>
> > The paper introduces new terminology and a combination of ideas from optimal transport, information geometry, and generative modeling, which means the exposition is quite dense; readers not already familiar with these backgrounds might struggle with clarity in places.
>
> We agree and aim to make the paper as accessible as possible. Currently, in Appendix C (Background), we briefly review notions of sufficiency, Wasserstein spaces, and stastical manifolds. In our revision, we will include additional sections on optimal transport and recent generative modelling approaches.
>
> > Additionally, the core engineering and methodological components are not always clearly described. For instance, when using a DDPM as the decoder, it appears that the model is trained by backpropagating through the diffusion model's inference process. This effectively makes the training simulation-based and computationally demanding.
>
> We apologize for the lack of clarity -- **we do not backpropagate through the entire diffusion inference procedure**, which, as the reviewer correctly points out, would be computationally  costly.
>
> Instead, we optimize the encoder parameters through the standard conditional diffusion loss, not its full sampling chain. More generally, **the divergence $\mathfrak{d}$ in the GDE loss function will always just be the standard training objective for the conditional generator.** In many ways this simplicity is the appeal of the approach: we do not need to make any modifications to the generator or its training, just condition on the latent representation from the distribution encoder and backpropogate through both networks.
>
> We will explicitly specify this training procedure in our revised manuscript (see proposed revision of Sec. 2 in response to reviewer N5oL).
>
> > On line 76, shouldn't the notation be $S_{\cdot, k}$ to be consistent with the notation introduced earlier?
>
> Thank you for highlighting this. Here, we were referring to the multiset constructed by duplicating each element of the sample set $k$ times. In our revised manuscript, we will more clearly refer to this as
>
> $$S_m^{\\{k\\}} := \bigcup_{i=1}^k S_m$$
>
> > When using a DDPM as a decoder [...] do you backpropagate through the entire diffusion inference procedure?
>
> We do not backpropagate through the inference procedure (see response above).
>
> > Could you provide a more detailed explanation of how the chosen architectures are modified or constrained to achieve distributional invariance?
>
> The key architectural choice to maintain distributional invariance is in the pooling function used by the encoder. In Appendix D.2., we enumerate pooling functions which satisfy the requirements of distributional invariance. In our implementations we have opted for sample mean pooling. However, we show that sample median, and general M/Z estimators will also lead to distributional invariance -- while sum pooling does not (see Appendix Proposition 4).
>
> > L70, it would be helpful to clearly define the input and output domains of the encoder and decoder
>
> Thanks for this suggestion. We have implemented it in our revised Sec. 2 (same content rewritten to increase clarity) which can be found in our response to reviewer N5oL.
>
> > Section 3 is somewhat difficult to follow [...] the explanation—particularly for continuous labels—should be clarified
>
> Thank you for this suggestion. We will  provide pseudocode for sampling with continuous labels in the revised appendix. Our proposed text is as follows:
>
> ---
>
> **Sampling sets from datasets with continuous labels**
>
> To construct sets from datasets where the label information is continuous, we can sample sets from label neighborhoods (rather than simply grouping by label as in the discrete case). Below, we provide example code for such a sampling approach
>
> ```
> # Inputs:
> # D = {(x_k, y_k)}_k : dataset with continuous labels y_k
> # N : number of sets to construct
> # m : set size
>
> def construct_sets_from_continuous_labels(D, N, m):
>     sets = []
>     for i in range(N)
>
>         # 1. sample target label y_star (e.g. uniformly over label range)
>         y_star = sample_target_label(D)
>
>         # 2. compute similarity weights (using kernel of choice)
>         weights = [exp(-d(y_k, y_star)**2 / (2 * sigma**2)) for _, y_k in D]
>
>         # 3. normalize weights
>         total = sum(weights)
>         weights = [w / total for w in weights]
>
>         # 4. sample m indices according to weights
>         indices = sample_indices(weights, m)
>
>         # 5. form the set
>         S_i = [D[idx][0] for idx in indices]
>         sets.append(S_i)
>
>     return sets
> ```
>
> > Line 78 [...] doesn't seem particularly helpful since there is obviously a one-to-one correspondence between the sample set (S_m) and the empirical distribution (P_m).
>
> We apologize -- this seems to be a miscommunication which should be resolved by our revised $S^{\\{k\\}}$ notation above.
>
> While in many cases distinct $S_m$ correspond to distinct $P_m$, there are subtle yet important cases where multiple sample sets can result in the same empirical measure. Permutations $\pi (S)$ and proportional duplications $S^{\\{k\\}}$ (i.e., copying every data point $k$ times) result in the same empirical measure $P$. These two cases motivate our requirements for distributional invariance.
>
> > For the evaluation on the synthetic data benchmark, I'd recommend also computing the Sinkhorn divergence as a metric for the low-dimensional (i) Gaussian and (ii) Gaussian mixture settings.
>
> Thank you for this suggestion. We have now also computed the Sinkhorn divergence in the low-dimensional cases (see below). The results are qualitatively similar, showing that GDEs outperform alternatives.
>
> Sinkhorn divergences for synthetic data experiments from Table 1:
> | Method        | MVN  | GMM  |
> |---------------|------|------|
> | GDE           | 0.11 | 0.32 |
> | KME+DDPM      | 0.14 | 0.40 |
> | W2 Wormhole   | 0.21 | 0.83 |

---

### Note · Authors · 2025-08-13

We appreciate the thoughtful feedback shared by the reviewers. We are happy to see that all reviewers recognized the technical contributions of our work with high scores for significance, originality, and quality.

Reviewer comments largely fell into two categories: (1) regarding clarity and presentation, and (2) regarding baselines in the application sections. Overall, we believe we addressed all of the questions and concerns raised -- and this appears to be reflected in the positive reviewer assessments. Below, we summarize the most significant changes we proposed during the rebuttal period:

- In response to reviewer comments and suggestions related to clarity, we have rewritten Section 2 to improve readability without changing its content. The proposed text can be found in response to reviewer `N5oL`.
- We have also written a new section summarizing practical implementation details for GDEs, to be included in the main text or appendix depending on space constraints. The proposed text can be found in response to reviewer `eYba`.
- In the Applications section, we now include an additional strong baseline for scRNA-seq tasks (the VAE-based scVI) -- and in line with previous results, we show that GDEs offer stronger performance than existing approaches.
- We will include additional appendix sections: (1) on background related to generative modeling and optimal transport (2) on possible paths to extend our theoretical contributions to non-i.i.d. settings.

Once again, we thank the reviewers for their thoughtful consideration of our work.

---

### Decision · Program_Chairs · 2025-09-17

**Decision:**

Accept (poster)

**Comment:**

**Summary:** This paper proposes a new concept of Generative Distribution Embeddings (GDEs), extending autoencoders from single-point embeddings to distributional embeddings. The framework consists of an encoder, enforced by distributional invariance, which maps a sample set into a latent code, and a conditional generator that decodes the latent code back into the distribution. Importantly, GDE distances approximately recover the Wasserstein-2 distance between distributions.

**Strengths:** GDEs introduce a novel paradigm for embedding distributions rather than single datapoints, with theoretical guarantees linking latent geometry to Wasserstein distances. Empirically, the framework outperforms existing baselines across both synthetic benchmarks and diverse biological applications.

**Weaknesses:** The paper’s presentation is weak and could be improved for broader accessibility, as all reviewers noted clarity issues. Sections 1–3 lack sufficient references, and most of the proofs are loose and asymptotic, relying on intuition and high-level arguments with limited rigor.

**Review Process**: Reviewers initially raised concerns about clarity, novelty scope, alternative metrics, qualitative analyses, and interpretability. In the rebuttal, the authors addressed most of these concerns, and three reviewers raised their scores from (4, 4, 5, 3) to (5, 5, 5, 4).

**Recommendation**: This paper presents a novel and well-motivated concept of distributional embeddings, supported by theoretical arguments and validated through experiments. The work is both original and impactful, and with reviewers reaching consensus after rebuttal, I am happy to recommend acceptance. The authors are encouraged to improve the presentation and strengthen the proofs to ensure broader accessibility and clarity.